# The *Bartonella* autotransporter BafA activates the host VEGF pathway to drive angiogenesis

Kentaro Tsukamoto [1✉], Naoaki Shinzawa[2,9], Akito Kawai [1], Masahiro Suzuki [1], Hiroyasu Kidoya [3], Nobuyuki Takakura [3], Hisateru Yamaguchi [4,10], Toshiki Kameyama [5,11], Hidehito Inagaki [6,7], Hiroki Kurahashi[6,7], Yasuhiko Horiguchi [2] & Yohei Doi [1,8✉]

Pathogenic bacteria of the genus *Bartonella* can induce vasoproliferative lesions during infection. The underlying mechanisms are unclear, but involve secretion of an unidentified mitogenic factor. Here, we use functional transposon-mutant screening in *Bartonella henselae* to identify such factor as a pro-angiogenic autotransporter, called BafA. The passenger domain of BafA induces cell proliferation, tube formation and sprouting of microvessels, and drives angiogenesis in mice. BafA interacts with vascular endothelial growth factor (VEGF) receptor-2 and activates the downstream signaling pathway, suggesting that BafA functions as a VEGF analog. A BafA homolog from a related pathogen, *Bartonella quintana*, is also functional. Our work unveils the mechanistic basis of vasoproliferative lesions observed in bartonellosis, and we propose BafA as a key pathogenic factor contributing to bacterial spread and host adaptation.

[1] Department of Microbiology, Fujita Health University School of Medicine, Toyoake, Aichi 470-1192, Japan. [2] Department of Molecular Bacteriology, Research Institute for Microbial Diseases, Osaka University, Yamada-oka, Suita, Osaka 565-0871, Japan. [3] Department of Signal Transduction, Research Institute for Microbial Diseases, Osaka University, Yamada-oka, Suita, Osaka 565-0871, Japan. [4] Division of Biomedical Polymer Science, Institute for Comprehensive Medical Science, Fujita Health University, Toyoake, Aichi 470-1192, Japan. [5] Division of Gene Expression Mechanism, Institute for Comprehensive Medical Science, Fujita Health University, Toyoake, Aichi 470-1192, Japan. [6] Division of Molecular Genetics, Institute for Comprehensive Medical Science, Fujita Health University, Toyoake, Aichi 470-1192, Japan. [7] Genome and Transcriptome Analysis Center, Fujita Health University, Toyoake, Aichi 470-1192, Japan. [8] Division of Infectious Diseases, University of Pittsburgh School of Medicine, Pittsburgh, PA 15261, USA. [9] Present address: Department of Environmental Parasitology, Graduate School of Medical and Dental Sciences, Tokyo Medical and Dental University, Yushima, Bunkyo-ku, Tokyo 113-8510, Japan. [10] Present address: Department of Medical Technology, School of Nursing and Medical Care, Yokkaichi Nursing and Medical Care University, Yokkaichi, Mie 512-8045, Japan. [11] Present address: Department of Physiology, Fujita Health University School of Medicine, Toyoake, Aichi 470-1192, Japan. ✉email: tsuka-k@fujita-hu.ac.jp; yoheidoi@fujita-hu.ac.jp

The α-proteobacterial genus *Bartonella* are Gram-negative, hemotropic, vector-borne, facultative intracellular pathogens that infect a broad range of mammalian hosts, and some are associated with human bartonellosis. *B. bacilliformis*, the pathogen of Carrión's disease, causes Oroya fever in the acute phase and verruga peruana in the chronic phase of illness[1]. *B. quintana*, the culprit of trench fever, and *B. henselae* which causes cat-scratch disease in immunocompetent persons can both cause bacillary angiomatosis in immunocompromised hosts[2]. In addition, the recently described *B. elizabethae* can also cause bacillary angiomatosis in HIV-positive patients[3]. Both bacillary angiomatosis and verruga peruana are characterized by the formation of hemangioma at skin lesions caused by abnormal endothelial cell proliferation[4,5]. The formation of such vasoproliferative lesions has been regarded as a hallmark of *Bartonella* infection not found in other pathogenic bacteria[6,7]. Vascular endothelial cells are an important target for bartonellae in their mammalian hosts, but whether *Bartonella*-triggered vasoproliferation represents a bacterial strategy to expand habitat in the host or only a coincidence is presently unclear. The molecular basis of *Bartonella*-triggered vasoproliferation, which is best studied in *B. henselae*, depends on multiple factors that are crucial for the infection cycle. *B. henselae* produces seven *Bartonella* effector proteins (BepA–G), and injects them through the VirB/D4 type IV secretion system (T4SS) into endothelial cells[8]. BepA is required for inhibition of apoptosis and contributes to the formation of vascular tumor[9]. Furthermore, *B. henselae* can infect macrophages or epithelial cells and promote the production of vascular endothelial growth factor (VEGF), thereby contributing to *B. henselae*-induced angiogenesis in a paracrine manner[10,11]. The VEGF production from the cells relies on the presence of pilus-like appendage comprised of *Bartonella* adhesin A (BadA) on the cell surface of *B. henselae*[12–14]. Notably, on the other hand, *B. henselae* can facilitate proliferation of endothelial cells without direct contact, indicating that the bacteria secrete mitogen(s)[15,16]. However, the identity of this mitogen is unknown.

Here, we successfully identify an autotransporter as the *Bartonella*-produced mitogen. The secreted passenger domain of the autotransporter exhibits pro-angiogenic activity in addition to its mitogenic property. As the cellular/molecular mechanism, the passenger domain recognizes VEGF receptor-2 (VEGFR2) on the cell surface and upregulates downstream mitogen-activated protein kinase (MAPK) pathway that results in the proliferation of vascular endothelial cells. Moreover, the homologs of the autotransporter are widely distributed in other *Bartonella* species, and one of them, the homolog in *B. quintana* also show the mitogenic activity. Therefore, the identified autotransporters uniquely observed in *Bartonella* are defined as a crucial virulence determinant and considered a new member of VEGF superfamily.

## Results

**Identification of BafA.** *B. henselae* promotes endothelial cell proliferation without exogenously supplemented growth factors. To assess the cell proliferation triggered by *B. henselae*, a cell proliferation assay was developed to quantify the number of human umbilical vein endothelial cells (HUVECs) by using Opera Phenix High-Content Screen (HCS) system. The HCS system enables counting of the cell numbers directly from fluorescence images. The cell numbers increased approximately 1.5-fold after 3 days of *B. henselae* infection at the multiplicity of infection (MOI) of 30 (Fig. 1a). In order to identify the *B. henselae* gene essential for their cell proliferative ability, transposon-based random mutants of *B. henselae* strain ATCC49882 were generated and screened for the number of HUVECs (Supplementary Fig. 1). Of the 1090 transposants, 79 transposants

showed reduced cell proliferation by primary screening (Supplementary Fig. 2). After secondary screening (Supplementary Fig. 3), two transposants, 623-125 and 804-29, were obtained that reproducibly exhibited cell proliferative capability that was comparable with uninfected cells (<110% of uninfected cells; Fig. 1b). These two transposants had little effect on cell proliferation of HUVEC even though the MOI increased by 100-fold (Fig. 1c). By inverse PCR followed by sequencing, both clones were found to contain the *mariner*-transposon at different positions in the same gene (locus tag: AYT27_RS02860) which encoded an autotransporter outer membrane β-barrel domain-containing protein (Fig. 1d). This protein was named BafA for *Bartonella* angiogenic factor A. By whole genome sequencing, no other transposon insertions were identified in the genomes of these two transposants. Since various clones showing different growth rates were obtained from the transposon library through this screening, we evaluated the growth rate and cell invasion ability of clones 623-125 and 804-29 (Supplementary Fig. 4). As a result, 623-125 showed no difference in the growth rate from wild-type (WT), but 804-29 exhibited lower growth rate than WT. On the contrary, cell invasive ability was elevated in 804-29 but not 623-125. Based on these findings, spontaneous mutation (s) unrelated to *bafA* may have affected the bacterial growth and cell invasiveness in 804-29, thus we used clone 623-125 as the *bafA*-disrupted strain in the subsequent experiments. Next, to evaluate whether *bafA* was essential for cell proliferative ability of *B. henselae*, clone 623-125 was complemented by *bafA*-carrying plasmid pBAF (623-125/pBAF). The presence of *bafA* restored the cell proliferative ability of the transposant, while the control vector did not (Fig. 1e). Previous studies have shown that *B. henselae* secretes one or more mitogenic factors that enhance endothelial cell proliferation[15,16]. Therefore, mitogenicity conferred by *bafA* was examined under conditions where HUVECs and the bacteria were co-cultured without direct contact (Fig. 1f). WT and *bafA*-complemented strains promoted cell growth, but the transposant with the control vector did not (Fig. 1g). These data indicate that *bafA* is essential for the mitogenic property of *B. henselae*.

**bafA modulates angiogenesis-related genes.** To further elucidate the effect of *bafA* on endothelial cells, global gene expression profiles of HUVECs in the presence of bacterial cells were examined by RNA sequencing (RNA-seq). The RNA-seq data were generated from HUVECs that were indirectly co-cultured with *B. henselae* strains as shown in Fig. 1f, and the genes with statistically significant expression changes were identified in comparison with bacteria-free cells. Expression changes were observed in 48 genes (40 upregulated and 8 downregulated) in co-cultures with *B. henselae* WT, while the expression levels of 11 genes (ten upregulated and one downregulated) were altered in co-cultures with the transposant strain 623-125. In the co-culture with the complemented strain 623-125/pBAF, differential expression was observed in 47 genes (33 upregulated and 14 downregulated) (Fig. 2a). The functions of differentially expressed genes (DEGs) were classified into inflammatory response, immune response, and response to lipopolysaccharide categories based on Gene Ontology (GO) across the strains tested. In contrast, genes classified into the angiogenesis category were enriched in co-cultures with the WT or complemented strain, but not the transposant strain (Fig. 2b–d). Most of the 78 DEGs exhibited high expression levels in co-cultures with the WT or complemented strain 623-125/pBAF compared with the transposant strain (Fig. 2e). Of note, 10 of 13 angiogenesis-related genes showed higher expression levels in the WT and complemented strains than in the transposant strain (Fig. 2f). These results

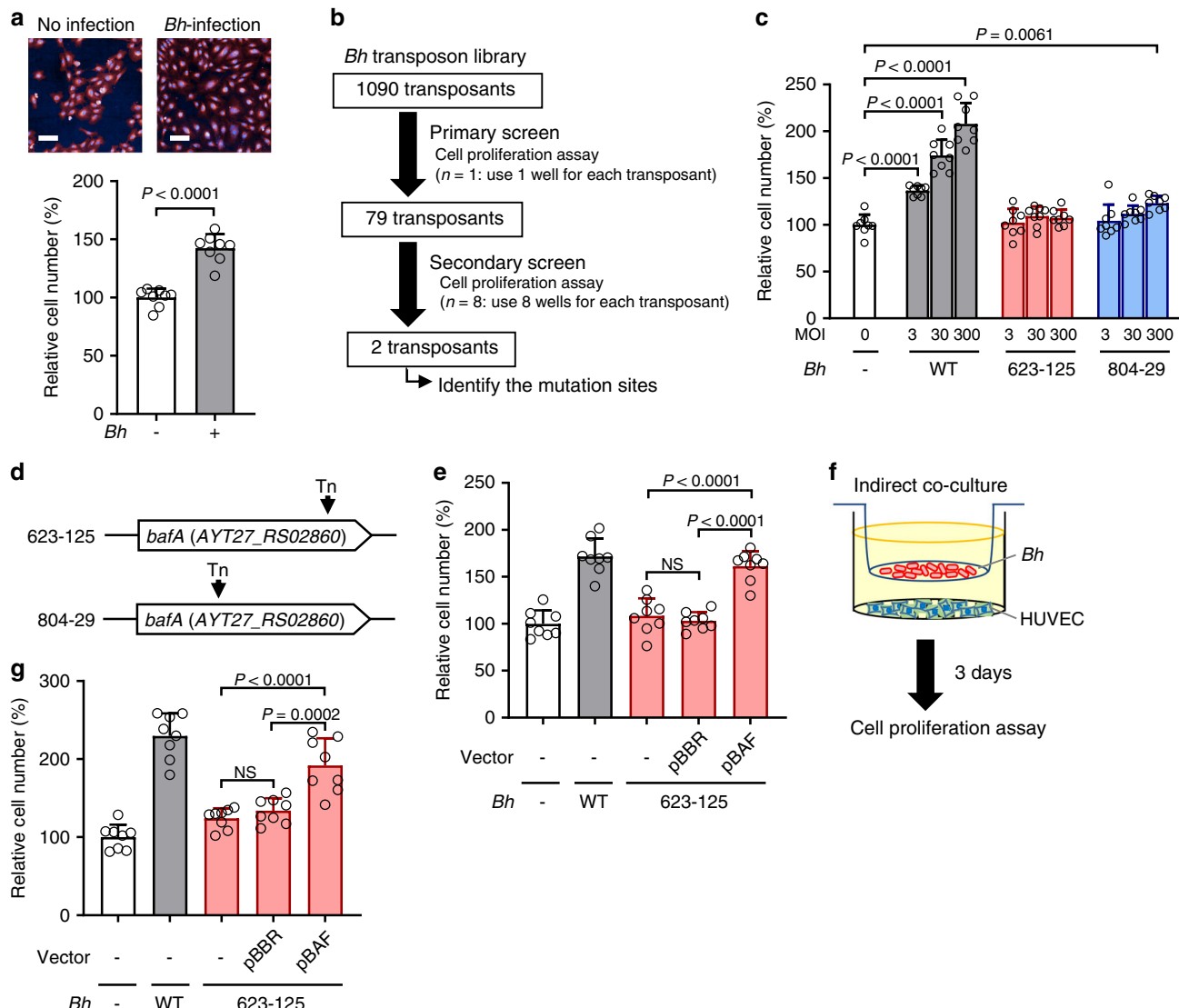

**Fig. 1 Identification of *B. henselae* gene essential in promoting vascular endothelial cell proliferation. a** *B. henselae* (*Bh*) promotes vascular endothelial cell proliferation. Human umbilical vein endothelial cells (HUVECs) were co-cultured with or without *Bh* at multiplicity of infection (MOI) of 30 for 3 days. Fluorescence images stained with CellMask (red) and Hoechst 33342 (blue) are shown. Scale bar = 100 μm (upper). The numbers of cells from the images were counted, and the values were normalized to the uninfected cells as 100% (lower). **b** Screening process for *Bh* transposants lacking cell proliferative ability. **c** Cell proliferative ability of *Bh*-WT and two transposants isolated from the screening. HUVECs were co-cultured with *Bh* cells (WT, 623-125, or 804-29) at MOI of 3, 30, or 300 for 3 days, and the relative cell numbers are shown. **d** Diagram of *bafA* (*AYT27_RS02860*) in two transposants, showing the locations of transposon insertion in 623-125 (at base pair position 646,986 in the *Bh* genome; NCBI accession no. NC_005956.1) and 804-29 (at position 645,770). **e** Ectopic expression of *bafA* restores *Bh*-induced cell proliferation. HUVECs were co-cultured with WT or 623-125 transposants complemented with *bafA* expression plasmid pBAF. The transposants without vector and with empty vector (pBBR) were used as negative controls. After 3-day incubation, relative cell numbers were counted. **f** Schematic representation of indirect co-culture of *Bh* and HUVECs. **g** *Bh* cells with *bafA* promote cell proliferation without direct contact. HUVECs were co-cultured indirectly with or without indicated *Bh* strains. After 3-day incubation, cell numbers were calculated. Bars are colored based on the strain: white for no infection, gray for WT, red for 623-125 and blue for 804-29 (**a**, **c**, **e**, **g**). Data are mean ± s.d. (*n* = 8 biological replicates). Statistical significance was determined using a two-tailed unpaired Student's *t* test (**a**), one-way ANOVA with Dunnett's multiple comparisons test (**c**), or with Tukey's multiple comparisons test (**e**, **g**). **$P$ < 0.01; ***$P$ < 0.001; NS not significant. All experiments were performed at least three times, and one of repeats is shown for each. Source data are provided as a Source Data file.

indicate that BafA production in *B. henselae* specifically increase the expression levels of angiogenesis-related genes in endothelial cells.

**BafA is a pro-angiogenic factor of *B. henselae*.** Based on BLAST searches, *bafA* encodes a putative autotransporter protein of type V secretion system, which translocates its passenger domain through a pore formed by its β-barrel domain. Some passenger domains of autotransporters are processed and released from the

outer membrane to serve as toxins, or to impair the host immune response[17–20]. Hence, the passenger domain of BafA was predicted to be secreted into the culture medium and promote endothelial cell proliferation. To test this hypothesis, we examined whether BafA could be detected in the culture supernatant of Strep-tagged BafA-expressing *B. henselae* (Supplementary Fig. 5a). Using immunoblot analysis, a band of ~53 kDa that reacted with the anti-Strep-tag II antibody was observed in the recovered fraction of the supernatant (Supplementary Fig. 5b). In

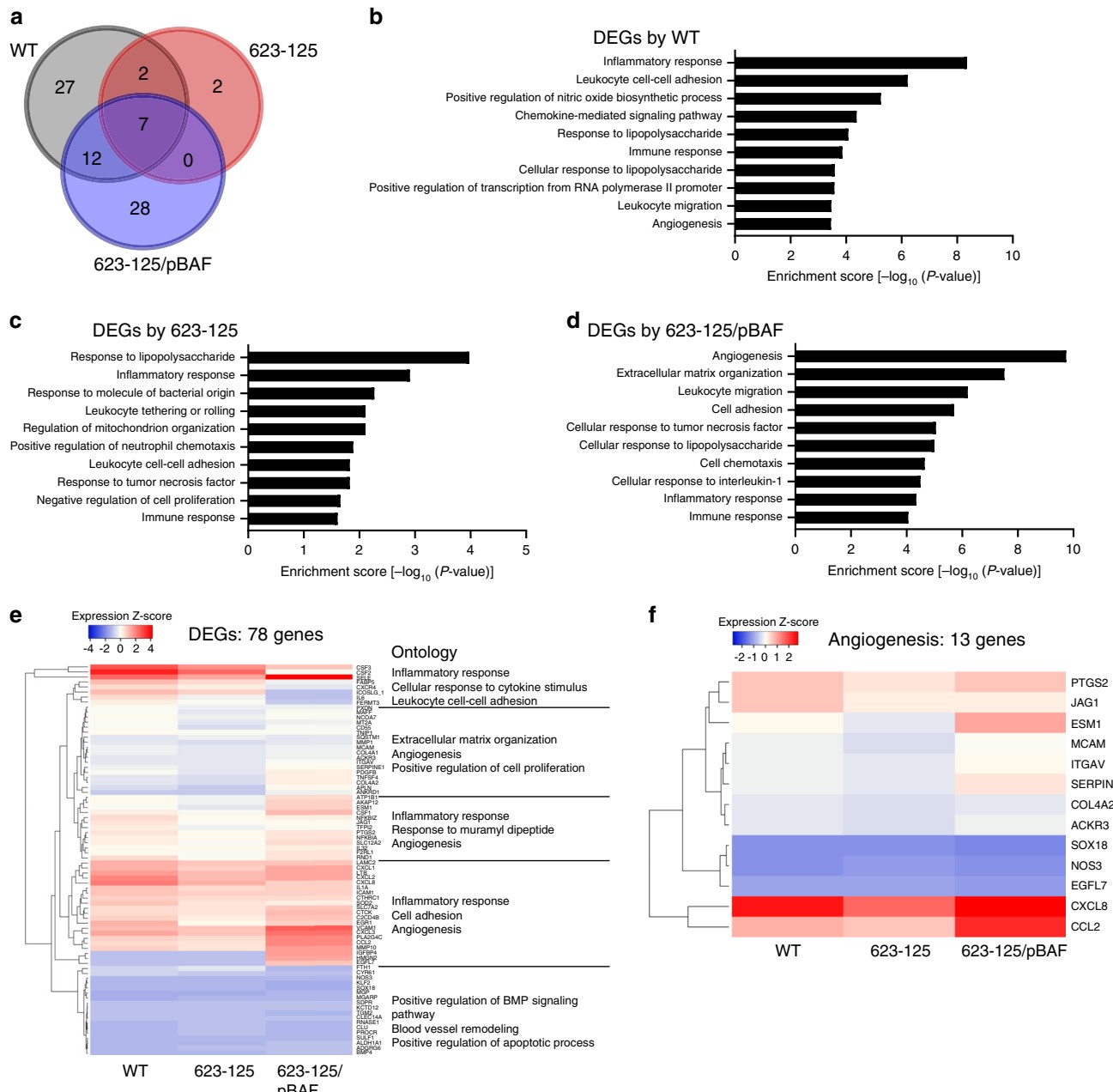

**Fig. 2 The *bafA* gene affects angiogenesis-related gene expression in endothelial cells. a** Venn diagram showing the overlap among differentially expressed genes (DEGs; fold change compared with uninfected HUVECs >1.5, three biological replicates) in HUVECs co-cultured with three *Bh* strains: WT (gray), 623-125 (red), or 623-125/pBAF (blue). **b–d** Enrichment of GO categories among DEGs in HUVECs co-cultured with *Bh*-WT (**b**), 623-125 (**c**), or 623-125/pBAF (**d**). **e** Heatmap representation of expression of all DEGs and selected enrichment GO terms for each cluster. **f** Heatmap of angiogenesis-related genes extracted from DEGs. Source data are provided as a Source Data file.

addition, nano-liquid chromatography (LC) mass spectrometry (MS)/MS detected the BafA peptides derived from the passenger domain without the β-barrel domain in this fraction (Supplementary Fig. 5c and Supplementary Table 1). These observations indicate that *B. henselae* is capable of releasing the BafA passenger domain extracellularly. Accordingly, we next generated the recombinant BafA passenger domain (BafA-PD, theoretically 53 kDa) and examined its mitogenic and pro-angiogenic activities. BafA-PD was constructed to contain homologous domains with two adhesin involved in diffuse adherence (AIDA) domains and a pertactin-like (PL)-passenger domain (Fig. 3a). Exogenously added BafA-PD increased the number of HUVECs in a dose-dependent manner, and the proliferation rates of BafA-PD-

treated cells were similar to those treated with VEGF (Fig. 3b). We then used three cell lines (HeLa 229, CHO-K1, and MRC-5) to evaluate the cell specificity of mitogenic activity of BafA-PD, among which no remarkable activity was observed (Supplementary Fig. 6). This proliferative activity of BafA-PD was completely blocked by the addition of anti-BafA polyclonal antibody (Fig. 3c). Moreover, the anti-BafA antibody also decreased *B. henselae*-induced cell proliferation, suggesting that BafA-PD may be the major mitogen secreted by *B. henselae*.

Next, we examined the pro-angiogenic activity of BafA-PD in a cell-based assay: a tube formation assay of HUVECs sandwiched between type I collagen gel in the presence or absence of BafA-PD (Fig. 3d–g). After 24 h culture, the cells formed few tube-like

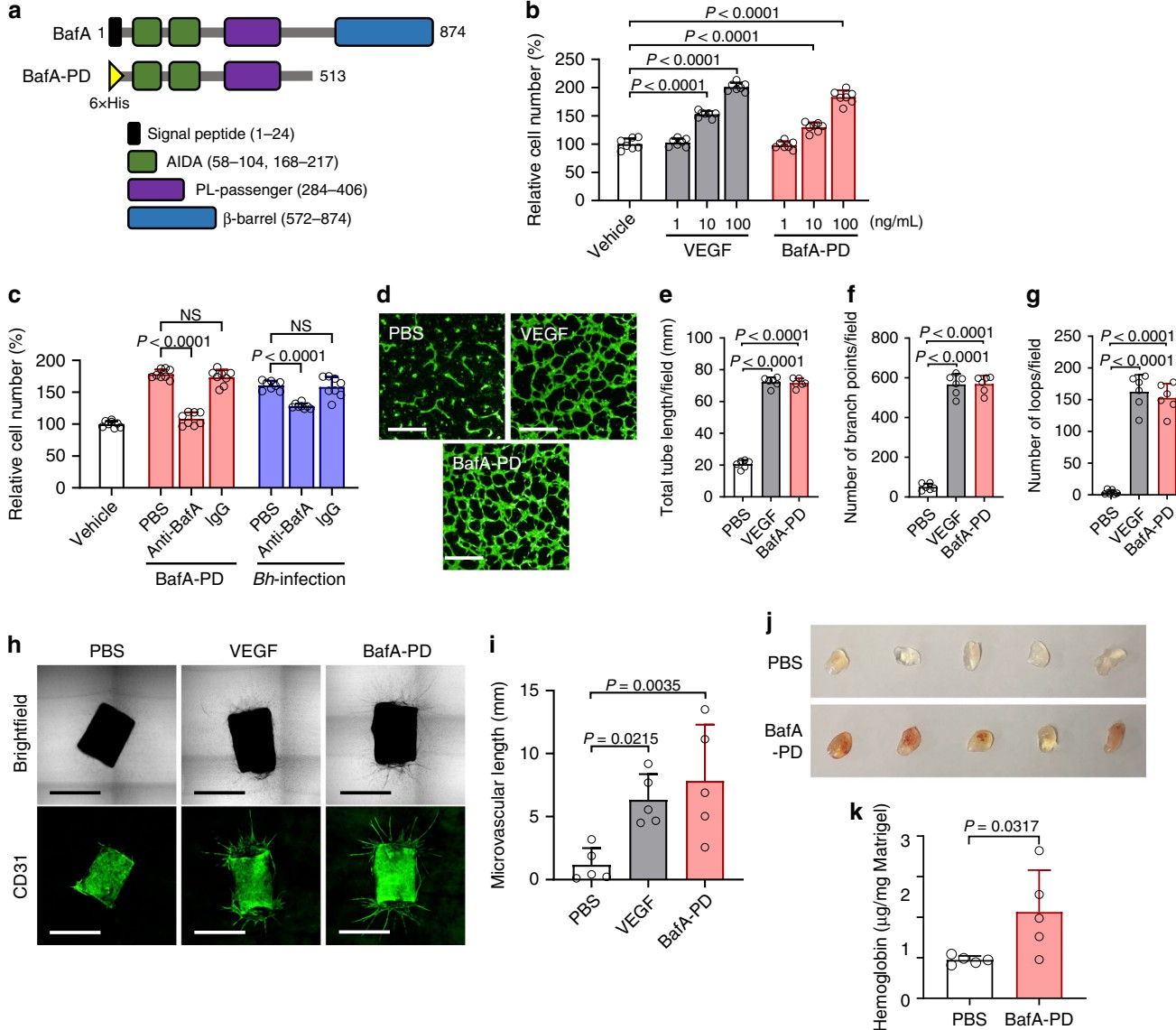

**Fig. 3 BafA passenger domain possesses angiogenic activity. a** Schematic representation of the full-length BafA autotransporter and the recombinant passenger domain (BafA-PD). His-tagged BafA-PD was constructed from the amino acid residues 25–513 of the full-length protein. AIDA, adhesin involved in diffuse adherence; PL-passenger, pertactin-like passenger domain; β-barrel, β-barrel domain. **b** Dose-dependent cell proliferation activity of BafA-PD (*n* = 7 biological replicates). **c** Effect of anti-BafA antibody on cell proliferation. HUVECs were treated with BafA-PD (100 ng/mL) or *Bh*-infection (MOI of 300) in the presence of anti-BafA antibody or normal rabbit IgG (*n* = 8 biological replicates). NS, not significant. **d–g** Tube formation assay. HUVECs were sandwiched in type I collagen gel and incubated with VEGF or BafA-PD for 24 h. Confocal images stained with calcein AM are shown. Scale bar = 500 μm. Similar results were obtained in three independent experiments (**d**). Total tube length (**e**), number of branch points (**f**), and number of loops (**g**) were calculated from the images (*n* = 6 biologically independent samples). **h, i** Aortic ring assay. Representative images of microvessels sprouting from mice aortic rings. The upper photographs show brightfield images and the lower photographs show fluorescence immunostaining images using anti-CD31 antibody. Five rings in each group were imaged, and one of them is shown. Scale bar = 1 mm. Similar results were obtained in three independent experiments (**h**). The microvascular lengths were measured from five fluorescence immunostaining images in each group (**i**) (*n* = 5 biological replicates). **j, k** Matrigel plug assay. Mice were subcutaneously injected with Matrigel containing PBS or BafA-PD. Representative photographs of Matrigel plugs excised 10 days post injection from five mice in each group (**j**). Hemoglobin content of excised Matrigel plugs (*n* = 5 biological replicates) (**k**). Bars are colored based on the treatment: white for vehicle, gray for VEGF, red for BafA-PD, and blue for *Bh*-infection (**b, c, e–g, i, k**). Data are mean ± s.d. Statistical significance was tested using one-way ANOVA with Dunnett's multiple comparisons (**b, c, e–g, i**) or two-tailed Mann–Whitney *U* test (**k**). Source data are provided as a Source Data file.

structures in the absence of VEGF and BafA-PD. In contrast, BafA-PD as well as VEGF markedly increased the formation of tube-like structures, indicating that BafA-PD could stimulate the angiogenic processes in endothelial cells. The aortic ring assay was then performed to examine whether BafA-PD could promote neo-vessel sprouting and branching in tissue culture[21,22]. As shown in Fig. 3h, i, VEGF and BafA-PD facilitated microvascular sprouting from the aortic rings. Finally, to assess the ability of BafA-PD to promote angiogenesis in vivo, the Matrigel plug assay was conducted[23]. Matrigel plugs supplemented with PBS or BafA-

PD were subcutaneously injected to mice, then the plugs were excised, photographed, and assayed for hemoglobin content 10 days after injection. Figure 3j shows a representative set of Matrigel plugs taken from a group of five mice. The levels of angiogenesis within the plugs were evaluated by the degree of red color in the plugs and hemoglobin content. Each Matrigel supplemented with PBS remained colorless, whereas most of the Matrigel plugs treated with BafA-PD turned yellow or red. The observed difference in the colors was confirmed by quantification of hemoglobin. BafA-PD treatment significantly increased the hemoglobin content compared with PBS treatment (Fig. 3k). These in vivo results, coupled with the in vitro and ex vivo observations, strongly suggest that BafA-PD is a pro-angiogenic factor.

**BafA induces angiogenesis through VEGFR2 signaling pathways**. The above-described results suggest that BafA promotes angiogenesis like VEGF. VEGF induces neovascularization primarily through VEGFR2, which is a major angiogenic receptor that plays a crucial role in both physiological and pathological conditions[24,25]. Specifically, phosphorylated VEGFR2 binds to phospholipase C-γ (PLCγ) and activates the MAPK/extracellular-signal-regulated kinase-1/2 (ERK1/2) pathway and proliferation of endothelial cells[26]. We therefore hypothesized involvement of the VEGFR2 signaling pathway in BafA-induced angiogenesis. As expected, BafA could induce phosphorylation of VEGFR2, MAPK kinase (MEK1/2), and ERK1/2 within HUVECs (Fig. 4a). The level of BafA-induced phosphorylation was comparable with that of VEGF-induced phosphorylation (Fig. 4b–d). On the other hand, stimulation of BafA had no substantial effect on Akt phosphorylation (Fig. 4a, e). We further evaluated whether the VEGFR2 signaling pathway could be activated by the production of BafA from bacterial cells using a cell-based infection model. The *bafA*-disrupted strain (623-125) did not induce VEGFR2 phosphorylation whereas WT and *bafA*-complemented strains did (623-125/pBAF) (Supplementary Fig. 7). These findings confirmed *bafA*-dependent regulation of VEGFR2-ERK1/2 pathway in *B. henselae*. To assess the role of VEGFR2-ERK1/2 signaling in mediating BafA-induced cell proliferation, HUVECs were pretreated with either U0126 (MEK1/2 inhibitor) or Ki8751 (VEGFR2 inhibitor) before BafA treatment (Fig. 4f–i). Pretreatment with U0126 completely inhibited BafA-induced phosphorylation of ERK1/2, but not VEGFR2. Ki8751 blocked VEGFR2 phosphorylation and reduced the amount of phosphorylated ERK1/2 within BafA-treated cells (Fig. 4f–h). The same results were observed within VEGF-treated cells. With basic fibroblast growth factor (bFGF)-treated cells, U0126 could abolish ERK1/2 phosphorylation, but Ki8751 did not influence phosphorylation of ERK1/2 likely due to differences in cell surface receptors. In addition, pretreatment with either U0126 or Ki8751 significantly reduced the number of BafA- or VEGF-treated cells (Fig. 4i). U0126 was also able to block bFGF-induced cell proliferation, but Ki8751 was not. These results indicate that BafA induces VEGFR2 signaling and drives cell proliferation. The effect of the VEGFR2 inhibitor Ki8751 on *B. henselae*-induced cell proliferation was then examined. A concentration of Ki8751 that does not affect uninfected cells markedly suppressed cell proliferation in *B. henselae*-infected cells (Fig. 4j), strongly suggesting that BafA recognizes VEGFR2 and upregulates the MAPK/ERK pathway. In further support of this hypothesis, immunoprecipitation assay revealed interaction between extracellularly added BafA-PD and VEGFR2 (Fig. 4k). On top of that, the presence of anti-VEGF neutralizing antibody had no effect on BafA-induced cell proliferation, while anti-VEGFR2 antibody remarkably inhibited mitogenic activity of BafA (Supplementary

Fig. 8). These results also indicate that BafA represents a novel VEGFR2 ligand whose antigenic property is distinct from VEGF. Collectively, these observations provide evidence that BafA plays a significant role as an agonist of VEGFR2, and that BafA-dependent VEGFR2 signaling is an essential step for *B. henselae*-mediated cell proliferation.

**BafA homolog in *B. quintana* also has mitogenic activity**. Like *B. henselae*, *B. quintana* is also known to induce remarkable vasoproliferation, which manifests clinically as bacillary angiomatosis[27,28]. Accordingly, we speculated that *B. quintana* might produce a mitogen that is similar to BafA. Previously reported genome analysis of *Bartonella* species have determined the homologous genes shared by *B. henselae* and *B. quintana*[29]. Of those, a gene encoding an autotransporter outer membrane β-barrel domain-containing protein in *B. quintana* (locus tag: BQ_RS02370) was identified as the likely *bafA* homolog. The passenger domain of BafA homolog in *B. quintana*, BafA$_{Bq}$-PD, consists of a 481-amino acid protein, which shares 59.7% sequence identity with the *B. henselae*-derived BafA-PD (Supplementary Fig. 9) and contains two AIDA and one PL-passenger domains (Fig. 5a). Extracellularly added BafA$_{Bq}$-PD showed cell proliferation ability to HUVECs only at concentrations of more than 500 ng/mL, indicating that BafA$_{Bq}$-PD has lower activity than *B. henselae*-derived BafA (Fig. 5b). Like BafA-PD, BafA$_{Bq}$-PD promoted tube-like formation of HUVECs (Fig. 5c–f). The findings suggest that *B. quintana* secretes a BafA homolog that may contribute to the formation of vasoproliferative lesions observed in bacillary angiomatosis. Furthermore, Protein BLAST search revealed that BafA-homologous proteins are widely distributed in *Bartonella* species but not other organisms, and that the sequences are distantly related with the present members of the VEGF family (Fig. 5g, Supplementary Fig. 10).

**Discussion**

*Bartonella* species are well known to induce vasoproliferation in humans upon infection, but the underlying mechanism has remained unclear. *Bartonella* type IV effector proteins have been shown to translocate directly into host cells, exert antiapoptotic activity and increase endothelial cell survival, but *B. henselae* mutants with disrupted VirB/D4 T4SS can still stimulate robust endothelial cell proliferation[30]. In contrast to the cell proliferation, employing in vitro angiogenesis assay of collagen gel-embedded HUVEC spheroids, *B. henselae*-induced capillary sprouting from the spheroids were seen in a T4SS, especially BepA, dependent manner; conversely, BepG, another effector of T4SS, potently inhibited sprouting[31]. These contradictions have not been resolved and also indicate that T4SS-independent pro-angiogenic determinant is involved in *Bartonella*-triggered angiogenesis. In addition, neither *virB* nor other T4SS-related genes are observed in the genome of *B. bacilliformis*, which is an ancestral lineage of *Bartonella* species, but even so this pathogen causes severe angiogenesis clinically[29]. The absence of VirB T4SS in *B. bacilliformis* also suggests that T4SS is not essential for pro-angiogenic capability of *Bartonella*. Of the other important pathogenicity factors, BadA has also been reported to possess pro-angiogenic activity. BadA is one of the trimeric auto-transporter adhesins which consists of over 3000 amino acid residues, and mediates the binding of *B. henselae* to endothelial cells by the formation of pilus-like structure on the surface of the outer membrane[14]. Expression of BadA also correlates with activation of hypoxia inducible factor-1 and subsequent secretion of pro-angiogenic cytokines (e.g., VEGF or interleukin-8) from epithelial cells[12,13]. The BadA-dependent VEGF secretion was only detected using HeLa 229 cells or *Bh*-infected THP-1

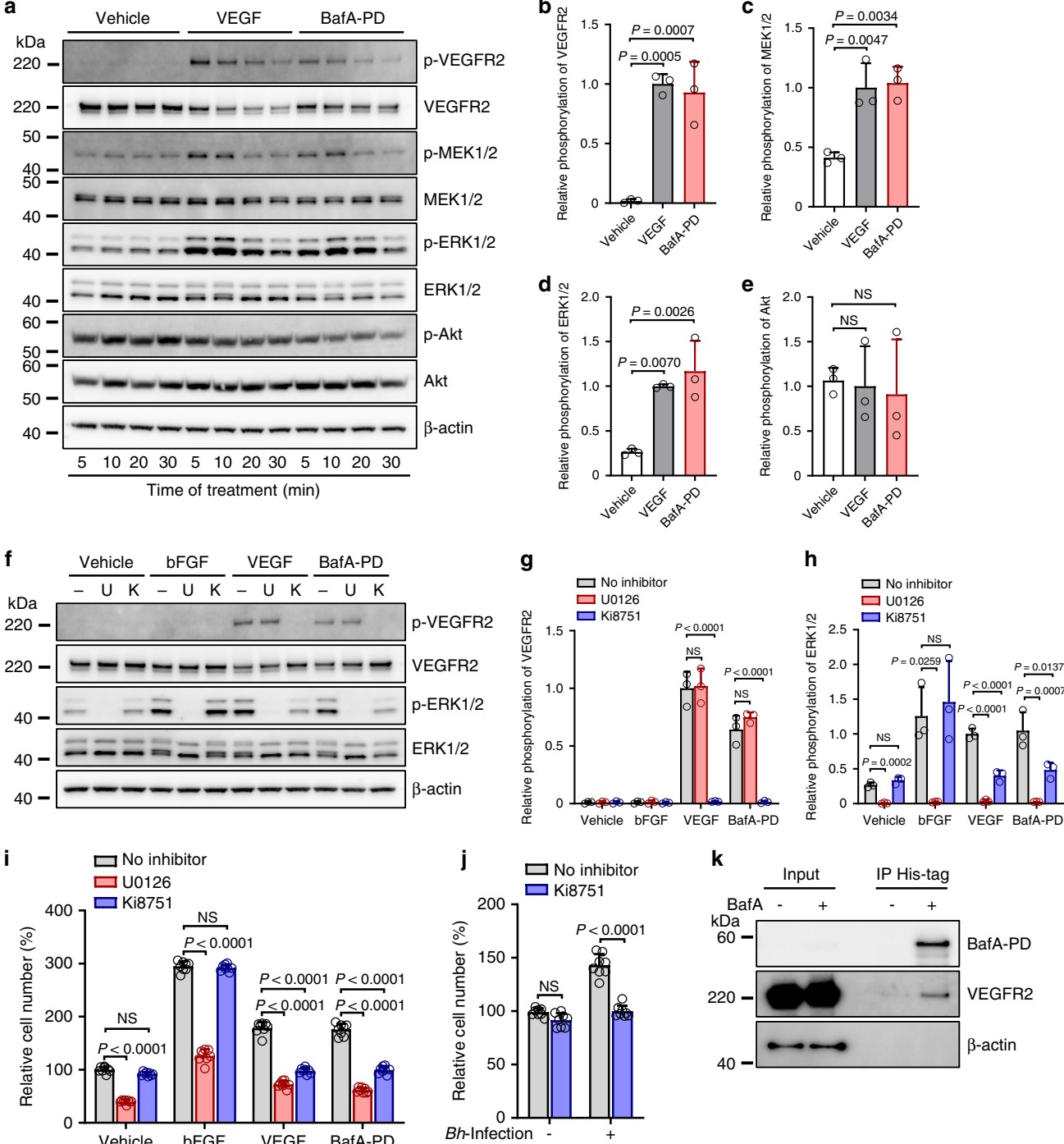

**Fig. 4 BafA upregulates VEGFR2-ERK signaling pathway. a–e** BafA leads to phosphorylation of VEGFR2, MEK1/2, and ERK1/2 but not Akt. HUVECs were stimulated with VEGF (20 ng/mL) or BafA-PD (100 ng/mL) for indicated time, then subjected to immunoblotting using antibody against phosphorylated (p-) or total VEGFR2, MEK1/2, ERK1/2, Akt, or β-actin (**a**). Quantification of p-VEGFR2 (**b**), p-MEK1/2 (**c**), p-ERK1/2 (**d**) and p-Akt (**e**), represented as the ratio of phosphorylated to total proteins at 10 min ($n = 3$ biological replicates). Bars are colored based on the treatment: white for vehicle, gray for VEGF and red for BafA-PD (**b–e**). **f–h** MEK1/2 and VEGFR2 inhibitors suppress BafA-induced phosphorylation. − no inhibitor, U U0126 (MEK1/2 inhibitor), K Ki8751 (VEGFR2 inhibitor). Immunoblots (**f**), quantification of p-VEGFR2 (**g**), and p-ERK (**h**) are shown ($n = 3$ biological replicates). **i** MEK1/2 and VEGFR2 inhibitors suppress BafA-induced cell proliferation. HUVECs were treated with each inhibitor before stimulation with bFGF, VEGF, or BafA-PD. Three days after treatment, the cell numbers were counted ($n = 8$ biological replicates). **j** VEGFR2 inhibitor suppresses *Bh*-induced cell proliferation. HUVECs were treated with Ki8751 before *Bh* infection. Three days after infection, the cell numbers were counted ($n = 8$ biological replicates). **k** Co-immunoprecipitation (IP) shows interaction between BafA and VEGFR2. HUVECs were treated with or without BafA-PD, then BafA-PD was immunoprecipitated with anti-His-mAb-magnetic beads. Immunoblotting was performed to detect BafA and VEGFR2. β-actin served as a loading control for inputs. Similar results were obtained in three independent experiments. Data are mean ± s.d. (**b–e**, **g–j**). Statistical significance was tested using one-way ANOVA with Dunnett's (**b–e**) or Tukey's (**g–j**) multiple comparisons test. Source data are provided as a Source Data file.

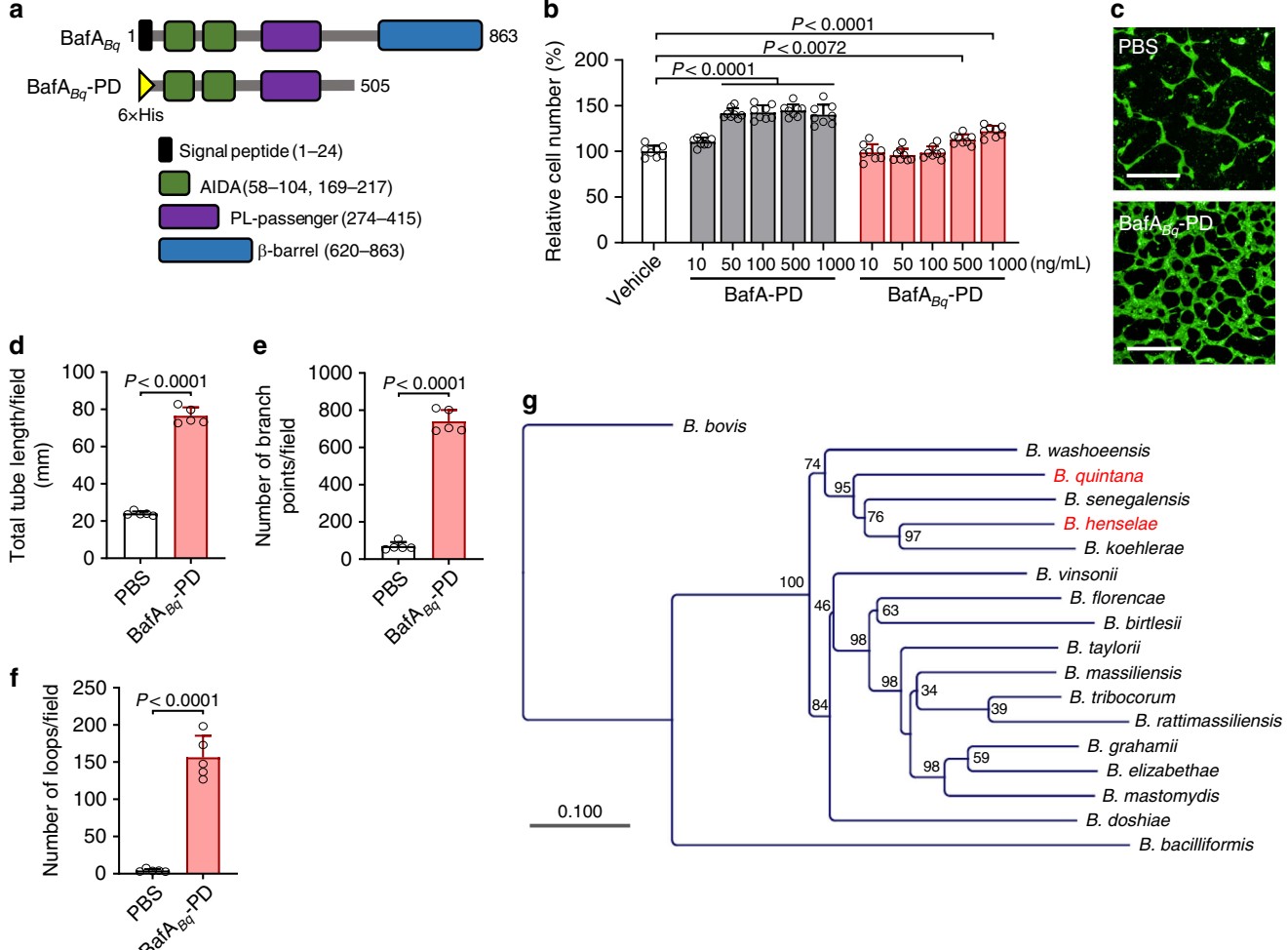

**Fig. 5 The BafA homolog from *B. quintana* possesses in vitro angiogenic activity. a** Schematic representation of *B. quintana*-derived full-length BafA (BafA$_{Bq}$) and the recombinant passenger domain (BafA$_{Bq}$-PD). The full-length BafA$_{Bq}$ has a similar domain structure with the *Bh*-derived BafA. BafA$_{Bq}$-PD was constructed as a His-tagged protein from the amino acid residues 25–505 of BafA$_{Bq}$-PD. **b** Comparison of cell proliferation activities of BafA$_{Bq}$-PD and BafA-PD ($n = 8$ biological replicates). **c**–**f** Tube formation assay. HUVECs sandwiched between collagen gels were incubated in the presence of PBS or BafA$_{Bq}$-PD (300 ng/mL) for 24 h. The tube formation was imaged (**c**), and the total tube length (**d**), number of branch points (**e**), and number of loops (**f**) were calculated as described above. Scale bar = 500 μm ($n = 6$ biological replicates). Bars are colored based on the treatment: white for vehicle or PBS, gray for BafA-PD, and red for BafA$_{Bq}$-PD (**b**–**e**). Data are mean ± s.d. (**b**, **d**–**f**). Statistical significance was tested using one-way ANOVA with Dunnett's multiple comparisons test (**b**) or two-tailed unpaired student's *t* test (**d**–**f**). **g** Phylogenetic tree of the BafA family proteins. The red letters indicate the species used in this study. Numbers at nodes indicate the percentage of bootstrap values. Source data are provided as a Source Data file.

macrophages[10], but no significant VEGF production was observed in *B. henselae*-infected HUVECs[11]. Therefore, both T4SS-dependent antiapoptotic activity to endothelial cells and BadA-dependent paracrine cytokine production may contribute to *Bartonella*-induced vasoproliferation. Indeed, the observation that proliferation of HUVECs occurs without direct contact between the bacteria and endothelial cells has suggested the presence of an extracellular mitogenic factor[15], but the identity of this mitogenic factor has remained elusive. As a candidate of secretory mitogen produced by *Bartonella*, the presence of GroEL chaperone in *B. bacilliformis* supernatant has been associated with its mitogenic activity against HUVECs[32]. This mitogenic activity was suppressed by the presence of anti-GroEL antiserum, and GroEL levels in the lysate correlated with HUVEC proliferation. However, anti-GroEL antiserum was not able to completely abrogate mitogenic activity, and significant mitogenic activity of recombinant GroEL was not observed regardless of dosage, suggesting that the role of GroEL might be indirect and other proliferative factors were important in the mitogenicity. Thus, these previous observations did not fully explain *Bartonella*'s mitogenicity.

We carried out this study to identify the *Bartonella* mitogen and characterize the signaling pathways involved in this process. Consequently, we found an autotransporter protein BafA as a key *Bartonella*-derived mitogenic factor. Typically, autotransporters, also known as bacterial type V secretion systems, consist of two functionally distinct domains: a secreted N-terminal passenger domain and a C-terminal β-barrel domain. The β-barrel domain is anchored in the Gram-negative outer membrane and translocates the passenger domain extracellularly. The passenger domains harbor the specific function or enzymatic activity as virulence factors, and their functions are extremely diverse ranging from adhesins to enzymes or toxins[33]. In some cases, the passenger, such as SPATE (serine protease autotransporters of Enterobacteriaceae) protease[34] or AIDA-I[35], can be proteolytically processed after exposure to the bacterial cell surface and released into the extracellular environment. The BafA passenger, like these autotransporter proteins, also appears to be released

through the channel formed by its β-barrel domain and act as an angiogenic factor. In contrast, BadA, one of the best characterized *Bartonella* autotransporter, consists of head, neck, extended coiled-coil stalk, and C-terminal membrane anchor domains[14]. In addition, variably expressed outer membrane proteins (Vomps), the BadA homologs in *B. quintana*, has the same domain structure as BadA[36,37]. It has been postulated that the stalk domain in these trimeric autotransporter adhesin, formed from an α-helical coiled-coil, is necessary for trimerization[38]. On the other hand, the passenger domain of BafA is predicted to only consist of a β-sheet structure and is not therefore expected to have coiled-coil stalk module. On top of that, the BadA head is composed of three domains: a YadA-like head repeat, a Trp ring, and a GIN domain[39], whereas BafA-PD does not conserve any of these motifs or domains. From these obvious differences between BafA and BadA (or Vomps), we speculate that BafA does not form a trimeric state seen with BadA. However, further structural analyses are needed to clarify whether BafA could multimerize.

In terms of the function of BafA, our results demonstrate that BafA of *B. henselae* recognizes VEGFR2 on the endothelial cell surface and activates the MAPK/ERK pathway, resulting in the promotion of cell proliferation and subsequent angiogenesis. Intriguingly, a previous study described that pre-infection of HUVECs by *B. henselae* interfered with the VEGF-induced responses (sprouting, proliferation, or migration) of the cells[40]. At the molecular level, the bacteria inhibited phosphorylation of VEGFR2 and subsequent downstream signaling in a T4SS-dependent manner. This previous observation appears to be inconsistent with our results. One possible explanation for this discrepancy is different variants of *B. henselae* used in these studies. It is known that at least two variants exist in the *B. henselae* ATCC49882 strain[41]. One variant (var-1) expresses the VirB/D4 T4SS but fails to express the adhesin BadA, and are not capable of enhancing VEGF secretion from infected HeLa cells. The other variant (var-2) displays symmetrically opposite properties (T4SS-negative, BadA- and VEGF induction positive). The authors of the previous article used T4SS positive var-1, infection by which reduces the VEGF response, whereas the strain used in the present study expresses BadA but few T4SS-related proteins (Supplementary Table 2) and may exhibit a property different from var-1. In fact, 24 h after bacterial infection, phosphorylation of VEGFR2 was observed. That is, different strains or variants of *B. henselae* may utilize multiple pathogenic factors (e.g., BafA, BadA, or T4SS) to modulate the pro-angiogenic response of endothelial cells.

In mammals, the VEGF family is composed of five members: VEGF-A, B, C, D, and placental growth factor[25,42]. In addition, similar proteins exist in parapoxvirus (VEGF-E)[43] and snake venom (VEGF-F)[44]. On the other hand, no VEGF analogs produced by bacteria have been identified. Here, we present a series of evidence that *Bartonella*, causative agents of bacillary angiomatosis, produce VEGF-like virulence factors. While many functional autotransporters have been reported from other Gram-negative bacteria, none is known to act as a VEGF analog to promote angiogenesis. Since both *B. henselae*- and *B. quintana*-derived BafA facilitate endothelial cell proliferation and tube formation, the BafA family autotransporters may be consequential virulence factors of many pathogenic *Bartonella* species, and the BafA family proteins may represent new members of the VEGF superfamily. To our knowledge, this is the first report to identify bacteria-derived VEGF-like proteins. The BafA family proteins are widely conserved among many *Bartonella* species, but amino acid identity among them are not high (19.0–73.8%). Wagner et al. recently proposed that the remarkable degree of host adaptation of *Bartonella* species to their mammalian hosts could be attributed to the T4SS and its effector proteins[45]. Based

on this hypothesis, the variety of BafA family proteins may also contribute to the specific host adaptation and pathogenicity of each *Bartonella* species. Unlike *B. henselae* and *B. quintana*, which form vasoproliferative lesions almost exclusively in immunocompromised patients, *B. bacilliformis* cause the most striking and sometimes mortal cases of vasoproliferation in immunocompetent individuals. Based on these clinical observations and genetic divergence, it is tempting to speculate that the predicted BafA homolog of *B. bacilliformis* may harbor more potent pro-angiogenic activity than that of *B. henselae* or *B. quintana*. Generally, in reservoir hosts, *Bartonella* species (except for *B. bacilliformis*) exhibit neither their pathogenicity nor inducible vasoproliferation. Despite this, the endothelium has been considered to be still important for establishment of infection even in reservoir hosts, and it has long been proposed as the putative replicative niche for the bacteria as they invade the erythrocytes, referred to as "blood-seeding niche"[46]. Further studies on BafA production in the reservoir hosts may explain the reason for the differences in host adaptation or susceptibility toward bartonellae.

As for the conserved domain structure, BafA consists of two AIDA and one PL-passenger domains, a structure that is reported in a number of autotransporter adhesins in Gram-negative bacteria. Autotransporter adhesins are a major group of proteins that play important roles in bacterial pathogenesis[20]. They allow bacteria to attach to host cells, aggregate with other bacteria and form biofilms. The ancestral proteins of BafA might have originally functioned as adhesins, but BafA reported here recognizes VEGFR2 and behaves as its agonist. BafA production favors the survival of the facultative intracellular bacterium *Bartonella* by allowing it to escape host immune systems. Recently, endothelial cells were shown to be intrinsically defective in xenophagy, which is a special form of the intracellular bulk degradation process known as autophagy, resulting in insufficient elimination of bacteria[47]. We postulate that the vascular endothelial cells form a protected growth environment for *Bartonella* species, and they target endothelial cells as a strategy for host adaptation. Indeed, in the cells, *B. henselae* can persist as bacterial aggregates within a unique intracellular structure called invasome[48]. The invasome formation strictly depends on VirB/D4 T4SS, which suggests a role for this process in the invasion or colonization of the bacteria[30]. On the other hand, electron microscopic data of biopsy specimens in the bacillary angiomatosis lesions suggest that these bacteria are located primarily extracellularly[49]. This previous observation is in agreement with the role BafA plays as a robust extracellular signaling molecule stimulating the proliferation of endothelial cells. Along with previously reported VirB/D4 T4SS, *Bartonella* effector proteins, and a trimeric autotransporter adhesin BadA, BafA is an essential pathogenic determinant for their infection cycle.

In conclusion, the present study represents a major advance in the understanding of the pathophysiology of bartonellosis. The BafA-triggered angiogenesis plays a central role in the formation of vasoproliferative lesions in *Bartonella* infection. In addition, *B. henselae* and *B. quintana* have been reported to cause infectious endocarditis, and the number of cases of blood culture-negative endocarditis that have been attributed to *Bartonella* has steadily increased[50]. Because BafA is secreted extracellularly, it may be a potential target for development of diagnostic and therapeutic strategies for *Bartonella*-derived endocarditis. Meanwhile, angiogenesis is an essential physiological process in the fetal development, wound healing, and female reproductive cycle[51] that is regulated by VEGF and its receptors. A number of growth factors regulate tissue repair, and their clinical application is currently explored[52]. In this context, BafA may serve as an attractive template for

engineering of novel VEGFs for potential use in regenerative medicine.

## Methods

**Reagents and antibodies**. MEK inhibitor U0126 was purchased from Promega. Ki8751, an inhibitor of the kinase activity of VEGFR2, was obtained from Cayman Chemical. Protease Inhibitor Cocktail (#25955) and Phosphatase Inhibitor Cocktail (#07574) were from Nacalai Tesque. Recombinant human $VEGF_{165}$ and bFGF were from Sigma-Aldrich. Purified rat anti-mouse CD31 (#553370, diluted at a ratio of 1:500) was from BD Pharmingen. The antibody against VEGFR2 (#2479, at 1:1000), p-VEGFR2 (#2478, at 1:1000), MEK1/2 (#8727, at 1:1000), p-MEK1/2 (#9154, at 1:1000), ERK1/2 (#4695, at 1:1000), p-ERK1/2 (#4370, at 1:2000), Akt (#4691, at 1:1000), p-Akt (#4060, at 1:1000), and β-actin (#4970, at 1:1000) were from Cell Signaling Technology. Purified mouse anti-Strep-tag II monoclonal antibody (#M211-3, at 1:1000) was from MBL. Human anti-VEGF (BEV-ACIZUMAB BS Intravenous Infusion 100 mg [Pfizer], at 3 μg/mL) was from Pfizer. Human anti-VEGFR2 (Cyramza Injection, at 3 μg/mL) was from Eli Lilly Japan. HRP-conjugated donkey anti-rabbit IgG (#711-035-152, at 1:4000) and HRP-conjugated goat anti-mouse IgG (#115-035-062, at 1:4000) were from Jackson ImmunoResearch. Alexa 488-conjugated goat anti-rat IgG (#A11006, at 1:500) was from Invitrogen. To prepare polyclonal antibody against BafA-PD, a Japanese White rabbit (strain; JW; 10-week-old female; Kitayama Labes Co., Ltd.) was immunized with BafA-PD by EveBioscience Ltd. Polyclonal antibodies (anti-BafA) were then purified by using rProtein A Sepharose Fast Flow (GE Healthcare).

**Bacterial strains and growth condition**. B. henselae strain Houston-1 (ATCC49882) and B. quintana strain 90-268 (ATCC51694) were purchased from American Type Culture Collection. The B. henselae strain we used was confirmed to produce BadA (Supplementary Table 2), but it remains unclear whether this BadA is surface-exposed and functional. B. henselae was grown on Columbia agar with 5% defibrinated sheep blood (CSB) at 37 °C in a humidified atmosphere and 5% $CO_2$ for 3–7 days. When required, kanamycin, trimethoprim, or ofloxacin was used at a final concentration of 25, 10, or 0.5 μg/mL, respectively. B. quintana was grown on heart infusion agar (Becton Dickinson) with 5% defibrinated rabbit blood at 37 °C in a humidified atmosphere and 5% $CO_2$ for 3–5 days. Escherichia coli strains were grown in Luria-Bertani (LB) agar or liquid medium (Becton Dickinson). When required, kanamycin was used at a final concentration of 25 μg/mL. For trimethoprim-resistant transformants, E. coli strains were grown in Mueller-Hinton agar (Becton Dickinson) or liquid medium containing 25 μg/mL of trimethoprim.

**Cell culture**. HUVECs were purchased from PromoCell. The cells were maintained in EGM-2 basal medium with SupplementPack (complete EGM, PromoCell) at 37 °C in a humidified atmosphere and 5% $CO_2$. Human cervical cancer HeLa 229, human lung fibroblast MRC-5, and chinse hamster ovary CHO-K1 cells were obtained from Japanese Collection of Research Bioresources cell bank. The cells were maintained in DMEM supplemented with 10% fetal bovine serum (FBS, Biowest) (for HeLa 229 and MRC-5; Thermo Scientific) or Ham's F-12 medium/10% FBS (for CHO-K1; Thermo Scientific) at 37 °C in a humidified atmosphere and 5% $CO_2$.

**Plasmid construction**. Primers used for plasmid construction are listed in Supplementary Table 3. pCVD442 was provided by Akio Abe and Asaomi Kuwae, Kitasato University. pNH3503 was provided by Hiroki Nagai, Gifu University. pCACTUS-Tp was provided by Asaomi Kuwae, Kitasato University. The plasmid pMariK used for the delivery of the mariner minitransposon containing a kanamycin-resistant cassette was constructed in two sequential steps. R6K ori plasmid pCVD442[53] was digested with XbaI and BstPI. A mariner minitransposon containing a kanamycin resistance gene was amplified by standard PCR with vector pNH3503[54] as the template and a combination of primers mariner-F and mariner-R and inserted into the 3.8 kb fragment of digested pCVD442 using In-Fusion HD cloning kit (Clontech). The resultant plasmid was designated pR6K-mariner. The rpsL gene and mariner transposase-encoding gene were amplified with pNH3503 as the template and a combination of primers rpsL-F and TPase-R and inserted into BstPI-digested pR6K-mariner. The resultant plasmid was designated pMariK. The plasmid pBAF was used for complementation of bafA in B. henselae and generated as follows. The broad host range expression vector with a trimethoprim resistance gene (TpR) was generated by replacement of an antibiotic resistance gene of the shuttle vector pBBR1MC-2[55]. TpR was amplified with a combination of primers TpR-F and TpR-R using pCACTUS-Tp[56] as the template. The amplified TpR DNA was inserted into the NcoI/BglII sites of pBBR1MCS-2, resulting in plasmid pBBR1-TpR. For constitutive expression of BafA protein, the entire bafA gene including the putative promotor site was amplified by PCR with primers RS02720-Prom-Fw and pBBR-RS02720-Rv using the B. henselae genomic DNA as the template. The PCR products were inserted into the BamHI/EcoRV sites of pBBR1-TpR. The constructed plasmid pBAF was introduced into a transposon-integrated mutant of B. henselae by biparental conjugation using the donor E. coli strain SM-10 as described in the section "Transposon mutagenesis". The bafA-complemented B. henselae strain was grown and maintained on CSB agar containing kanamycin

and trimethoprim. For the expression of Strep-tagged BafA in B. henselae, the region from putative promotor site to the signal sequence was amplified with primers RS02720-Prom-Fw and bafA-SP-Rv. In addition, the remaining bafA gene was amplified with primers pBBR-RS02720-Rv and strep-bafA-Fw including Strep-tag-coding sequence. These two fragments were also inserted into digested pBBR1-TpR using In-Fusion HD cloning kit. The resultant plasmid was named pSBAF. The constructed plasmid pSBAF was introduced into a transposon-integrated mutant of B. henselae by biparental conjugation using the donor E. coli strain SM-10. The Strep-tagged BafA-expressed B. henselae strain (623-125/pSBAF) was grown and maintained on CSB agar containing kanamycin and trimethoprim. For preparation of BafA-PD and $BafA_{Bq}$-PD proteins, the plasmids pET-28b-BH513 and pET-28b-BQ505 were generated as follows. The nucleotides encoding the 25–513th amino acids of BafA or the 25–505th amino acids of $BafA_{Bq}$ were amplified with primer sets NheI-Bh-Fw/SalI-Bh513-Rv or NheI-Bq-Fw/SalI-Bq505-Rv using B. henselae or B. quintana genomic DNA as the templates, respectively. The PCR products were digested with NheI/SalI and inserted into the compatible sites of pET-28b.

**Transposon mutagenesis**. To generate transposon insertions in B. henselae by biparental conjugation, the donor strain E. coli SM-10 carrying pMariK was grown overnight in LB medium containing kanamycin. The overnight culture was washed with 10 mM magnesium sulfate once and resuspend in an appropriate volume of heart infusion broth (Becton Dickinson) to an optical density at 600 nm ($OD_{600}$) of 1.0. The recipient strain B. henselae ATCC49882 was grown on CSB agar for 4 days, collected from plates, and suspended with an appropriate volume of heart infusion broth to an $OD_{600}$ of 8.0. B. henselae and donor E. coli suspensions were mixed at a ratio of 10:1, dropped onto CSB agar, dried for 30 min, then incubated for 7 h at 37 °C in a humidified atmosphere and 5% $CO_2$. The conjugated cells were collected, plated onto CSB agar containing 0.5 μg/mL of ofloxacin, and incubated for 15 h to remove the donor E. coli. Finally, the bacterial cells from the plate were suspended in heart infusion broth, spread onto CSB agar containing ofloxacin and kanamycin, and incubated for 10–20 days in a humidified atmosphere and 5% $CO_2$. The transposon-mutagenized B. henselae colonies were subjected to cell proliferation assay to screen for transposants lacking cell proliferation ability. For determination of insertion sites in such transposants, inverse PCR was applied. Briefly, genomic DNA was purified from 7-day-old cultures on CSB agar using PureLink genomic DNA mini kit (Invitrogen). The genomic DNA was digested with NlaIII which cuts only at one site within the transposon, and circularized by T4 DNA ligase (Promega). The digested and circularized genomic DNA was used as the template for PCR with Mari1 and Mari4 primers. The PCR products were separated by agarose gel electrophoresis, purified, and subjected to direct sequencing using Mari1 and Mari4 primers. The obtained sequences were aligned to the B. henselae genome (NCBI accession no. NC_005956.1) to identify the transposon insertion site. The deduced amino acid sequence of the identified gene was analyzed using SignalP 4.0 server[57] and NCBI's Conserved Domain Database[58,59]. In addition, in order to confirm the absence of transposon insertion at other sites, draft genome sequences of two transposants (clones 623-125 and 804-29) were generated. Genomic DNA was extracted and purified using PureLink genomic DNA mini kit. The MiSeq libraries for DNA sequencing were prepared using the Nextera XT DNA Library Prep Kit (Illumina). Sequencing was performed on an Illumina MiSeq with the Miseq reagent kit ver. 3 (600 cycles, Illumina), which generated 2,693,412 reads for 623-125 and 2,102,242 reads for 804-29, respectively. A Nanopore library was prepared using a 1D Ligation Sequencing Kit (SQK-LSK-109; Nanopore) and Native Barcoding Expansion (EXP-NBD104; Nanopore). Libraries were sequenced on a MinION using flow cell R9.4. The fast5 output was base-called using Guppy base caller and de-multiplexed using Guppy barcoder, and sequences which were 8 kbp or more were selected. The sequencing outputs were 90 Mbp for 623-125 (the predicted genome coverage depth was 45×) and 290 Mbp for 804-29 (the predicted genome coverage depth was 145×), respectively. Hybrid assembly was performed using Unicycler v. 0.4.8 under the default settings[60].

**Bacterial growth and invasion into HUVECs**. Bacteria (B. henselae WT, transposants 623-125, and 804-29, $OD_{600} = 0.05$, 100 μL) were plated on CSB agar plates to evaluate the bacterial growth. After the indicated days, bacteria were harvested from the agar surface by extensive washing with 1 mL of PBS. The $OD_{600}$ of the bacterial suspensions were measured in a JASCO V-630BIO spectropolarimeter (JASCO Corp.).

A gentamicin protection assay was used to calculate the number of intracellular bacteria. HUVECs seeded in a 24-well plate were infected with B. henselae WT or 623-125 (MOI = 200) for 24 h at 37 °C in a humidified atmosphere and 5% $CO_2$. After the infection, gentamicin sulfate (200 μg/mL) was added for 2 h to kill extracellular bacteria. The cells were then washed with Medium 199 (M199, Gibco) supplemented with 10% FBS three times and lysed by incubation for 15 min with 1% saponin. Cell lysates including intracellular bacteria were resuspended with 1000-fold volume of PBS. The numbers of bacteria were determined by plating on CSB agar.

**Expression and purification of recombinant BafA proteins**. E. coli BL21 (DE3) transformed with pET-28b-BH513 or pET-28b-BQ505 was grown at 18 °C in LB

medium containing kanamycin until the culture reached an $OD_{600}$ of 0.7. Protein expression was induced by adding 20 μM IPTG and incubation was continued overnight at 18 °C. The *E. coli* cells were harvested and resuspended in 25 mM Tris-HCl, pH 7.5, 500 mM NaCl, and 25 mM imidazole. After disruption of the *E. coli* cells by sonication, the lysates were obtained by centrifugation and filtration. Purification of BafA-PD was conducted through three chromatography steps: Ni-Sepharose affinity chromatography, gel filtration, and cation exchange chromatography. In Ni-Sepharose chromatography, BafA-PD bound to resin was washed with 25 mM Tris-HCl, pH 7.5, 500 mM NaCl, and 40 mM imidazole, 5 mM ATP, 10 mM $MgCl_2$, and 10% glycerol to remove contaminating proteins, and then eluted with 25 mM Tris-HCl, pH 7.5, 500 mM NaCl, and 300 mM imidazole. After concentrating the eluate by membrane ultrafiltration, the sample was subjected to gel filtration with a Superdex 200 column (GE healthcare) and the cell proliferation activity of each fraction was checked. Finally, the fractions possessing activity were purified by cation exchange chromatography using RESOURCE S column (GE healthcare). For purification of $BafA_{Bq}$-PD, two chromatography steps, Ni-Sepharose affinity chromatography, and gel filtration, were performed as described above. The protein concentration of the purified BafA-PD was determined using Protein Assay BCA Kit (Nacalai Tesque) according to the manufacturer's instructions.

**Cell proliferation assay.** HUVECs (passage 5–9, 10,000 cells/$cm^2$) were plated onto gelatin-coated 96-well plates with complete EGM. After 3 h of incubation, the cells were washed twice with M199/10% FBS and infected with indicated MOI of *B. henselae*. For evaluation of cell proliferation activity of BafA, the cells in M199/10% FBS were exposed to indicated concentrations of BafA. When examining the effect of inhibitors or antibodies, U0126 (10 μM), Ki8751 (50 nM), anti-BafA antibody (15 μg/mL), anti-VEGF (Bevacizumab, Pfizer, 3 μg/mL), anti-VEGFR2 (Ramucirumab, Eli Lilly Japan, 3 μg/mL), or normal human IgG (Fujifilm Wako, 3 μg/mL) were treated for 30 min (U0126 and antibodies) or 2 h (Ki8751) at 37 °C prior to infection with *B. henselae* or stimulation with BafA-PD. Three days after infection or BafA treatment, the cells were subjected to CellMask Deep Red plasma membrane dye (1:3000, Invitrogen) and NucBlue Live ReadyProbes Reagent (Invitrogen) for 30 min at 37 °C in a humidified atmosphere and 5% $CO_2$. The cells were then fixed with 4% paraformaldehyde for 15 min at room temperature and washed with PBS three times. Plates were imaged on an Opera Phenix HCS System (PerkinElmer) using the confocal setting with 5 or 20× air objective. Cell numbers were measured from images of 4 or 25 fields in the well using Harmony 4.5 software (PerkinElmer). To test the cell specificity for mitogenic activity of BafA, three cell lines: HeLa 229 (18,000 cells/$cm^2$), CHO-K1 (10,000 cells/$cm^2$), and MRC-5 (6000 cells/$cm^2$) were plated onto 96-well plates with the respective maintenance medium. After 4 h of incubation, the media were replaced by M199/10% FBS containing PBS, bFGF (20 ng/mL), VEGF (20 ng/mL), or BafA-PD (100 ng/mL). Three days later, the cells were stained, imaged and counted in the same manner as described above.

**Indirect co-culture of HUVECs and *B. henselae*.** To co-culture HUVECs with *B. henselae* without direct contact, MilliCell 24-well cell culture device (Millipore) was used to separate them. *B. henselae* cells cultured on CSB agar for 3 days were collected and suspended into M199/10% FBS to an $OD_{600}$ of 0.045. One hundred microliter of the suspended *B. henselae* cells were then added to filter plate wells and inserted into a receiver plate where HUVECs were seeded at a density of 15,000 cells/well in M199/10% FBS. After 3 days in co-culture, the MilliCell filter plate was removed, and the proliferation of HUVECs was assessed as described under "Cell proliferation assay".

**RNA-seq.** HUVECs co-cultured indirectly with or without *B. henselae* WT, 623-125 or 623-125/pBAF strains were cultured in M199/10% FBS for 48 h using MilliCell culture device as described above. Total RNA from HUVECs was then extracted and purified using ReliaPrep RNA Cell Miniprep System (Promega) according to the manufacturer's instructions. RNA concentrations were determined with Qubit RNA HS Assay Kit (Invitrogen), and integrity of RNA was confirmed as RIN score > 8 using an Agilent 2100 Bioanalyzer and RNA 6000 Nano Kit (Agilent). PolyA+ RNAs were purified from total RNA (200 ng) using NEBNext Poly(A) mRNA Magnetic Isolation Module (New England Biolabs). RNA-seq libraries were then constructed using the NEBNext Ultra RNA Library Prep Kit for Illumina (New England Biolabs) according to the kit's instructions. The libraries were sequenced with 125 bp single-end reads for each sample and three biological replicates per sample using an Illumina HiSeq 1500 at the Genome and Transcriptome Analysis Center of Fujita Health University. The bcl2fastq 1.8.4 software was used for base-calling, and CLC Genomic Workbench 8.5.2 software was used for quality trimming of raw sequences, alignment of trimmed reads to the human reference genome (GRCH38/hg38), and statistical analysis of DEGs (cut-offs: false discovery rate of <0.05 and $log_2$ fold change of >0.58 or <−0.58). GO analysis was performed with DAVID 6.8 (https://david.ncifcrf.gov/).

**Detection of BafA from culture supernatant.** *B. henselae* 623-125/pSBAF was cultured in M199/10% FBS containing 10 μg/mL of trimethoprim for 48 h at 37 °C

in a humidified atmosphere and 5% $CO_2$. The culture was centrifuged to remove bacterial cells, and the supernatant was then loaded onto Strep-Tactin Superflow resin column (IBA GmbH). The column was washed with 100 mM Tris-HCl, pH 8.0, 150 mM NaCl, and 1 mM EDTA, and eluted with 100 mM Tris-HCl, pH 8.0, 150 mM NaCl, 1 mM EDTA, and 2.5 mM desthiobiotin. The eluted fractions were subjected to SDS-PAGE followed by immunoblotting using anti-Strep-tag II monoclonal antibody. The proteins were visualized using HRP-conjugated anti-mouse antibody and SuperSignal West Dura or Femto Substrate (Thermo Scientific). The proteins included in the eluates were also identified using nano-LC–MS/MS.

**Nano-LC–MS/MS.** For detection of BafA from culture supernatant, the Strep-Tactin eluates described above were used for nano-LC–MS/MS analysis. Meanwhile, to confirm expression of BadA and T4SS in the WT strain used in this study, the bacterial cells of *B. henselae* strain ATCC49882 were harvested from 4-day cultures on CSB agar. The bacterial cells were then lysed with Cell Lysis Buffer in Pierce Mass Spec Sample Prep Kit (Thermo Scientific) and sonicated. The Strep-Tactin eluates and the whole cell extracts were reduced with 10 mM dithiothreitol (DTT) at 50 °C for 45 min and then alkylated with 50 mM iodoacetamide at 25 °C for 20 min. Alkylated samples were digested with lysyl endopeptidase (Thermo Scientific) at a 1:100 enzyme/protein ratio for 2 h at 37 °C, followed by trypsin (Thermo Scientific) at a 1:50 ratio for 16 h at 37 °C. Peptides were desalted with a MonoSpin C18 Column (GL Sciences), acidified with formic acid and analyzed by nano-LC–MS/MS. The LC conditions and the MS acquisition conditions were determined according to the previous study[61]. The peptides were loaded onto the LC system (EASY-nLC 1000; Thermo Scientific) equipped with a trap column (Acclaim PepMap 100 C18 LC column, 3 μm, 75 μm ID × 20 mm; Thermo Scientific), equilibrated with 0.1% formic acid, and eluted with a linear acetonitrile gradient (0–35%) in 0.1% formic acid at a flow rate of 300 nL/min. The eluates were loaded and separated on the column (EASY-Spray C18 LC column, 3 μm, 75 μm ID × 150 mm; Thermo Scientific) with a spray voltage of 2 kV (Ion Transfer Tube temperature: 275 °C). The peptide ions were detected using MS (Orbitrap Fusion ETD MS; Thermo Fisher Scientific) in the data-dependent acquisition mode with the Xcalibur software (version 4.0; Thermo Scientific). Full-scan mass spectra were acquired in the MS over 375–1500 *m/z* with resolution of 120,000. The most intense precursor ions were selected for collision-induced fragmentation in the linear ion trap at normalized collision energy of 35%. Dynamic exclusion was employed within 60 s to prevent repetitive selection of peptides. The MS/MS searches were performed using MASCOT (Version 2.6.1, Matrix Science) and SEQUEST HT search algorithms against the SwissProt and TrEMBL *Bartonella henselae* protein databases (v2017-10-25) using Proteome Discoverer 2.2 (Ver. 2.2.0.388; Thermo Scientific). The workflow for both algorithms included spectrum files RC, spectrum selector, MASCOT, SEQUEST HT search nodes, percolator, ptmRS, and minor feature detector nodes. Oxidation of methionine was set as a variable modification and carbamidomethylation of cysteine was set as a fixed modification. Mass tolerances in MS and MS/MS were set at 10 ppm and 0.6 Da, respectively. Trypsin was specified as protease and a maximum of two missed cleavages were allowed. Target-decoy database searches used for calculation of false discovery rate (FDR) and for peptide identification FDR was set at 1%.

**Endothelial cell tube formation assay.** In vitro endothelial tube formation assays were performed using collagen gel[62]. Collagen gel was prepared by mixing Cell-matrix type I collagen solution with MEM and reconstitution buffer (Nitta Gelatin). HUVECs ($7 \times 10^5$ cells/well in a 48-well plate) were cultured between two layers of collagen gel in the presence of PBS (control), VEGF (20 ng/mL), BafA-PD (100 ng/mL), or $BafA_{Bq}$-PD (1 μg/mL). EGM/2% FBS containing PBS, VEGF, BafA-PD, or $BafA_{Bq}$-PD were added to each well, then 24 h cultures were stained with 1 μM of calcein AM (Dojindo Molecular Technologies, Inc., Japan) at 37 °C for an hour. After washing with HBSS twice, fluorescence images were collected using Opera Phenix. The total tube length, the numbers of branch points, and the numbers of loops of tube-like structure in 24 fields per group were quantified using Harmony 4.5.

**Mice.** All animal experiments were approved by the Institutional Animal Care and Use Committee of the Fujita Health University, and carried out according to the Regulations for the Management of Laboratory Animals at Fujita Health University. C57BL/6J mice were purchased from Japan SLC. Mice were housed under 12 light/12 dark cycle, ambient temperatures of 23 ± 2 °C with 55 ± 10% humidity.

**Aortic ring assay.** The descending thoracic aortas were isolated from 8-week-old C57BL/6J mice. Aortic rings were placed between two layers of type I collagen gel with PBS, VEGF (30 ng/mL) or BafA-PD (200 ng/mL) in a 48-well plate. To the sandwiched rings, HuMedia-EG2 (Kurabo, Japan) containing PBS, VEGF, or BafA-PD was added and cultured for 6 days at 37 °C in a humidified atmosphere and 5% $CO_2$. For quantification of angiogenesis, the cultured aortic rings were immunostained with an anti-mouse CD31 mAb, followed by Alexa 488-conjugated secondary antibody. Following immunostaining, fluorescence images were collected using Opera Phenix, and the lengths of sprouting microvessels were calculated using Harmony 4.5.

**Matrigel plug assay**. The Matrigel plug assay was performed to evaluate the angiogenic response in vivo. Matrigel Growth Factor Reduced (0.45 mL, Corning) mixed with heparin (10 U/mL) and PBS or BafA-PD (500 ng/mL) were subcutaneously injected into the flanks of 8-week-old C57BL/6J mice. On day 10 post injection, mice were sacrificed, Matrigel plugs were removed, and blood vessel formation was evaluated by quantification of hemoglobin. The hemoglobin assay was performed as follows: Matrigel plugs were soaked in an equal volume of RBC lysis buffer (Sigma-Aldrich) and incubated overnight on ice. Drabkin's solution (Sigma-Aldrich) containing 0.3% Brij-35 was added to the samples and incubated for 30 min at room temperature. The hemoglobin concentration was determined by comparing the spectral absorption at 540 nm of the sample with that of hemoglobin standards (Sigma-Aldrich).

**Immunoblotting**. For pathway analysis of VEGF, HUVECs (passage 5–9, 20,000 cells/cm$^2$) were plated onto gelatin-coated 12-well plates with complete EGM. After 24 h of incubation, the medium was replaced with M199/10% FBS. The cells were starved for 3 h in the medium and then stimulated with or without VEGF (20 ng/mL) or BafA-PD (100 ng/mL) for the indicated time at 37 °C. When examining the effect of inhibitors, U0126 (10 μM) or Ki8751 (50 nM) was treated for 30 min (U0126) or 2 h (Ki8751) at 37 °C prior to stimulation of VEGF or BafA-PD. To verify the effect of *B. henselae*-infection on the VEGFR2 signaling pathway, the starved cells were infected with *B. henselae* WT, 623-125, or 623-125/pBAF (MOI = 500) for 24 h. The cells were harvested, and the cell lysates were obtained by ultrasonic treatment by using BIORUPTOR (Cosmo Bio, Tokyo, Japan) followed by centrifugation. Each cell lysates were then subjected to SDS-PAGE using NuPAGE 10 or 12% Bis-Tris gels and MOPS-SDS running buffer (Thermo Scientific), transferred onto PVDF membrane, and probed with antibodies to phosphorylated or total VEGFR2, MEK1/2, ERK1/2, and Akt using the iBlot 2 Gel transfer device and iBind Western System according to the manufacturer's instructions. The proteins were visualized using HRP-conjugated anti-rabbit antibody and SuperSignal West Dura or Femto Substrate (Thermo Scientific). To detect total proteins or internal control (β-actin), the blots were stripped with WB Stripping Solution (Nacalai Tesque) after detection of the phosphorylated protein and re-probed with another antibody. The density of the reactive bands was quantified using Multi Gauge version 3.0 (Fujifilm).

**Immunoprecipitation assay**. Eighty percent confluent HUVECs were starved for 4 h at 37 °C in the M199/1% FBS in a 100 mm dish, and treated with or without 500 ng/mL BafA-PD for 2 h at 20 °C. The cells were washed with cold Hank's balanced salt solution (HBSS) and lysed with 0.8 mL of lysis buffer (25 mM Tris-HCl, pH 7.4 containing 137 mM NaCl, 2.68 mM KCl, 1% Nonidet P-40, and 1% Protease Inhibitor Cocktail). After centrifugation, the resultant lysates were mixed with 20 μL of anti-His-tag mAb-Magnetic Agarose (MBL), and incubated for 1 h at 4 °C. The magnetic beads were then washed four times with the lysis buffer and mixed with NuPAGE LDS Sample Buffer containing 50 mM DTT. The samples were heated for 10 min at 70 °C and subjected to SDS-PAGE followed by immunoblotting.

**Bioinformatic analysis of BafA-homologous proteins**. The amino acid sequences of BafA autotransporters from *B. henselae* and *B. quintana* were aligned and compared by CLC Main Workbench 8.1.3 software. The secondary structures of the proteins were predicted by PSIPRED v4.0 on the PSIPRED Protein Analysis Workbench website (http://bioinf.cs.ucl.ac.uk/psipred/)[63]. To identify BafA homologs, the amino acid sequence of BafA-PD was queried against the NCBI Protein BLAST database of nonredundant protein sequences. The top hits from this search with an *E*-value threshold $1 \times 10^{-50}$ were selected as BafA-homologous proteins. The hits often included proteins from overlapping organisms or unidentified species; these hits were discarded except for the representative ones (Supplementary Table 4). The passenger domains of selected BafA-homologous proteins and VEGF family proteins were aligned and compared by CLC Main Workbench 8.1.3 software. From these pairwise comparisons, a cluster of sequences close to BafA-PD or BafA$_{Bq}$-PD (distance score of <0.7), as well as a predicted BafA homolog of *B. bacilliformis*, which is a particularly important pathogen for humans, were selected, and a phylogenetic tree was generated by maximum likelihood estimation using CLC Main Workbench. The robustness of phylogenetic inference was assessed by 1000 bootstrap replicates.

**Statistical analysis**. Data were expressed as the mean ± standard deviation (s.d.). Statistical significance was analyzed using two-tailed unpaired Student's *t* test, Mann–Whitney *U* test, one-way ANOVA with Dunnett's or Tukey's multiple-comparison test, and Kruskal–Wallis test with Dunn's multiple-comparison test, as indicated in the figure legends, with GraphPad Prism 8 (GraphPad Software). *P* values are defined as follows: *$P < 0.05$; **$P < 0.01$; ***$P < 0.001$; ****$P < 0.0001$. All in vitro experiments were conducted at least in triplicate to ensure reproducibility of the observations. Representative immunoblots and microscopy images are shown from at least three biologically independent replicates that showed similar results.

**Reporting summary**. Further information on research design is available in the Nature Research Reporting Summary linked to this article.

## Data availability

The RNA-seq raw data for each sample reported in this work have been deposited in the DDBJ Sequence Read Archive with the accession no. DRA009444. The draft genome sequences of *B. henselae* transposants 623-125 and 804-29 can be found in the DDBJ with the accession nos. BLJS01000000 and BLJT01000000, respectively. The MS raw datasets for the culture supernatant and the whole cell extracts have been deposited in the ProteomeXchange Consortium[64] via the jPOST[65] partner repository under data-set identifiers PXD017507 and PXD018354, respectively. Source data are provided with this paper.

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

## Acknowledgements
We thank Akio Abe and Asaomi Kuwae for providing pCVD442 and pCACTUS-Tp, Hiroki Nagai for pNH3503. We thank Suzu Katade for assistance with transposon-mutant screening, and Takao Tsuji, Yusuke Minato, and Keisuke Hitachi for their helpful comments. The imaging analysis using Opera Phenix HCS and the nano-LS–MS/MS analysis using Orbitrap Fusion ETD MS were performed at the Center for Joint Research Facilities Support of Fujita Health University. This work was supported by JSPS KAKENHI Grant Number JP19K07548 (K.T.), MEXT-Supported Program for the Strategic Research Foundation at Private Universities from the Ministry of Education, Culture, Sports, Science, and Technology of Japan (H. Kurahashi), and the Grant for Joint Research Project of the Research Institute for Microbial Diseases, Osaka University (K.T.).

## Author contributions
K.T. conceived the study and performed the majority of the laboratory experiments. N.S. constructed the necessary vectors. A.K. purified the recombinant proteins. M.S. performed the whole genome sequencing and bioinformatic analyses. H. Kidoya and N.T. designed ex vivo and in vivo angiogenesis assay. H.Y. performed MS analyses. T.K. prepared RNA-seq libraries. H.I. and H. Kurahashi performed Illumina sequencing for RNA-seq analyses. K.T., Y.H., and Y.D. outlined the study. K.T. and Y.D. wrote the initial draft paper. All authors contributed to the writing of the final paper.

## Competing interests
The authors declare no competing interests.
