## [Peer Review File · Nature Communications]

Reviewers' comments:

Reviewer #1 (Remarks to the Author):

This is a well-designed and important study that provides one of the first recent advances in some time on the unique angiogenic pathogenesis of *Bartonella*. The authors started from scratch by generating a transposon library in *B. henselae* and screening for loss of the proliferative activity for primary endothelial cells (HUVECs). They identified a novel trimeric autotransporter adhesin (BafA) as a factor that could induce HUVECs proliferation that is dependent upon VEGFR2 signaling. The manuscript is well-written and the experiments and resulting data are clearly presented. The authors should include evidence that the overall growth rate and fitness of the transposons mutants was not altered compared to the wild type as this would also impact cell proliferation. Additionally, since another trimeric autotransporter adhesin of *B. henselae* (BadA) has been reported to possess pro-angiogenic activity, this should be included in the discussion and compared to BafA.

Some additional specific suggestions and concerns follow.

Line 64: "inject" should read "injects"

Lines 69-70: The authors should cite a previous paper showing that VEGF production relies on the presence of BadA. BadA forms a pili-like appendage on the surface of the cell but is not technically a pilus.

Lines 75-76: Is the mitogenic activity specific for endothelial cells?

Lines 138-161 Could native (not-tagged) BafA-PD be detected in culture supernatants?

Lines 243-4: The authors may wish to also mention that *B. bacilliformis* was not observed to have the VirB type IV secretion system genes encoded in the genome. However, *B. bacilliformis* induces clinical angiogenesis - verruga peruana. Thus clinically severe angiogenesis resulting from *Bartonella* infection does not appear to require the type IV secretion system or its effectors.

Lines 245-9: The authors should compare BafA and BadA, a protein that has been previously reported to have pro-angiogenic properties. Since both proteins are trimeric autotransporter adhesions, is there any amino acid sequence similarity between the two proteins? Are there any common domains, Is there any structural similarity? Similarly, the BadA homologs in *B. quintana* (Vomps) should also be included in this discussion.

Line 341: Was the rabbit a specific strain (New Zealand) of white rabbit?

Line 346: Have the authors tested their *B. henselae* Houston-1 strain to determine if it expresses a functional badA gene? Some isolate3 have been reported to have a truncated version that might have been missed in the screen for cell proliferation. Additionally, repeated subculturing is thought to result in loss of badA expression.

Figure 1f and Figure 2a are similar and redundant. Deletion of Figure 2a is recommended.

Burt Anderson

Reviewer #2 (Remarks to the Author):

Tsukamoto et al., identify the *Bartonella* autotransporter BafA as a causative agent of cat scratch disease and angiomas. Using a high content screen for endothelial proliferation after infection with transposon-based random mutants of *B. henselae*, an autotransporter, denoted Baf A (*Bartonella* angiogenesis factor A) was identified as the causative agent. The authors go on to examine BafA gene regulation which clusters in Go terms such as "Angiogenesis". BafA induces endothelial cell responses in vitro, and endothelial cell outgrowth in aortic ring explants. Matrigel

plugs with BafA display increased levels of hemoglobin. Signal transduction analyses show activation of VEGFR2 and Erk1/2 signaling by BafA with close to the same efficiency as for VEGFA. Finally, the authors demonstrate that *B. Quintana* harbors a BafA homologue that stimulates endothelial cell proliferation. This is a careful and ambitious paper with convincing conclusions, which is likely to settle the question on the long-sought identity of causative agent for *B. henselae*-dependent angiomas.

1. Please show effects of BafA on endothelial cells in the presence of anti-VEGFA neutralizing antibodies.
2. What is the affinity of BafA for binding to VEGFR2?
3. Does BafA compete with VEGFA for binding to VEGFR2?
4. Does a *B. henselae* BafA mutant fail to induce angiomas?

Minor

5. Please cite the paper by Riess et al.

Bartonella adhesin A mediates a proangiogenic host cell response.

Riess T, Andersson SG, Lupas A, Schaller M, Schäfer A, Kyme P, Martin J, Wälzlein JH, Eehalt U, Lindroos H, Schirle M, Nordheim A, Autenrieth IB, Kempf VA.

J Exp Med. 2004 Nov 15;200(10):1267-78

6. Does BafA-induced gene regulation support induction of hypoxia-dependent expression pattern in *B. henselae* infected cells as described by Riess et al? Does BafA significantly induce expression of VEGFA? This should be clearly stated.
7. In the abstract, the authors state that BafA upregulates VEGFR2 which makes the impression that VEGFR2 expression levels are upregulated. However, the authors intention is likely to point out that BafA induces VEGFR2 signal transduction? Please rephrase.

Reviewer #3 (Remarks to the Author):

The study by Tsukamoto et al., presents the discovery and characterization of a novel pro-angiogenic factor secreted by the zoonotic pathogen *Bartonella*, which acts as an agonist of the VEGF-receptor 2 (VEGFR2). With an exhaustive set of experiments, the authors unambiguously demonstrate that the passenger domain of a bacterial autotransporter (named here BafA) is sufficient to trigger cell proliferation, eventually resulting in angiogenesis. Their data indicate that the passenger domain of BafA (BafA-PD), which is cleaved after translocation through the outer-membrane, constitute a novel secreted effector that is likely to play an important role in host colonization and/or adaptation of this stealthy pathogen.

At the molecular level, the authors show that BafA-PD interacts with VEGFR2 and triggers downstream signaling, thus acting as a VEGF-like molecule. Combining simple and complex in-vitro models, infections and use of recombinant proteins, the authors provide a very solid body of evidence, further confirmed using an elegant in-vivo assay.

This study is definitely an important contribution for the understanding of the pathogenesis of the pathogen *Bartonella*. Indeed, although the mitogenic properties of *B. henselae* have long been recognized, the underlying factor(s) remained elusive.

Moreover, the description of a bacterial VEGF-like secreted effector as a novel strategy for a pathogen to establish its infection niche will very likely catch interest to a broad scientific

audience.

Minor points

1. (General) One general issue I have on the overall report is a possible gap between the data on cell proliferation and those using angiogenesis as a read-out. On one hand, the authors present very convincing angiogenesis assays with purified BafA-PD. On the other hand, infection and cell signaling data only used cell proliferation as a read-out.

Although it is clear that cell proliferation can be used to some extent as a proxy for angiogenesis, the two processes are not fully congruent. To overcome that possible gap, I would suggest that the authors provide at least one infection experiment (with wild-type, BafA mutant and complemented strain) while assessing phosphorylation of the VEGF receptor-2 and the phosphorylation of ERK1/2. This would fully fill the gap. Alternatively, the conclusion could be somehow tuned to account for this gap. Further, it would be interesting to test whether the isolated BafA mutants are still able to induce VEGF secretion by HUVECs upon infection.

2. (General) The authors have used the *B. henselae* strain ATCC 49882 for their experiments. Importantly, it has been reported that at least two variants of the strain exist, likely as the result of in-vitro passaging (DOI: 10.1111/cmi.12070). One variant (var-2) expresses the trimeric autotransporter BadA but fail to express the VirB/D4 T4SS and to translocate T4SS effector in infected cells. This strain however strongly induce VEGF expression when infecting HUVECs. The other variant (var-1) displays symmetrically opposite properties (BadA negative, T4SS and effector translocation positive, no induction of VEGF). This has caused discrepancies in results obtained between different laboratories, and could be the underlying reason for the points raised in comments # 6 and #8. It would therefore be important to clarify the properties of the strain used for this study (BadA expression, T4SS expression, VEGF induction upon infection). Note that this point does not questions the pertinence of the BafA-related phenotype, but may be highly relevant for the interpretations of the global infection strategy of the studied pathogen.

3. (Line 69-70): Could the authors clarify the term "pili" used in the sentence "The VEGF production from the cells relies on the presence of *B. henselae* pili"? Do they refer to the trimeric autotransporter BadA?

4. (Lines 119-124). Could the authors indicate in the main text that data were generated by INDIRECT co-culture and that the statistically significant expression changes observed result from the comparison to uninfected cells? It is clearly stated in the figure legend, but adding it to the main text would increase clarity.

5. Regarding the RNA-seq analysis: could the authors provide a (supplementary) table with the genes displaying differential expression (presented in fig. 2) – including the normalized expression values for the three tested conditions? The Venn diagram presented in Fig. 2b actually only show a rather modest overlap between the wild-type and the complemented strain. Could this be due to the stringent statistical cut-off applied? A few word on the genes up-regulated in both wild-type and mutant strain could also be of general interest. Further, as the authors discovered that Baf-PD acts as a VEGF-like molecule, a comparison to the transcriptional response of HUVECs to VEGF treatment (from literature) would be appealing for the interpretation of the observed changes.

6. The authors convincingly demonstrate that BfaA-PD interacts with the VEGF receptor-2, acting as an agonist. They clearly show that Bfa-PD stimulates cell proliferation together with the activation of phosphorylation of VEGFR2 and ERK1/2. Interestingly, a previous report (DOI: 10.1111/j.1462-5822.2010.01545.x) assessed the effect of *B. henselae* (var-1) on VEGF signaling using infection of HUVECs and concluded that the pathogen actually inhibits VEGFR2 signaling (in VEGF-stimulated cells) in a T4SS-dependent manner. The authors showed that the same phosphorylation site activated by Bfa-PD (Tyr1175) is actually inhibited by their wild-type strain. This study should be cited and the discrepancies discussed (likely linked to point #2)

7. The authors propose BafA as a factor contributing to host adaptation (line 45). In that context, the authors could consider the following point for discussion. As stated in the introduction, vasoproliferative lesions seem to be restricted to the infection of human as an incidental host, and this mainly in case of immunosuppression (*B. henselae* and *B. elizabethae*). The most striking case of vasoproliferation described to date is the case of human infection by *B. baciliformis* (verruca peruana). It would make sense to include this species in the tree of BafA-homologues presented in fig. 5g.

The lack of report for vasoproliferative lesion in natural host (with the exception of *B. baciliformis*) does not exclude angiogenesis is a key requirement for the establishment of *Bartonella* infection in its natural host. Notably, the endothelium has already long been proposed as the putative replicative niche from which the bacteria invade the erythrocytes (blood seeding niche). That point could be addressed in the discussion.

8. Although the data presented unambiguously support the role of BafA in angiogenesis, this role was not tested in any infection model (only cell proliferation was). Intriguingly, a previous publication by Scheidegger et al., (DOI: 10.1111/j.1462-5822.2009.01313.x), which used infection of HUVECs-derived spheroids and sprouting as readout, concluded that angiogenesis was strongly promoted the VirB/D4 type IV secretion (T4SS) and the secreted effector BepA. However, they only observed "basal angiogenesis" in the absence of the T4SS. This is in clear contrast to experiments addressing cell proliferation, a process that was shown to be T4SS-independent. The study of Scheidegger et al., should be cited and the discrepancies briefly discussed (likely linked to the point raised in comment #2). It sure would be interesting to see the effect of a BafA mutant in an infection experiment with angiogenesis as read-out, but this may go beyond the scope of the present report.

Maxime Québatte, PhD

Responses to Reviewer's Comments

We are grateful to the reviewers' comments and useful suggestions that have helped us to improve our paper considerably. And also, we are pleased to note the favorable comments of the reviewers in their opening paragraphs.

*The comments provided by the reviewers are copied in blue italics. Our responses are shown in black roman type and the revised position (line number) in red. In addition, we have included in this letter the figures and tables upon which the responses to reviewer's comments are based (Appendixes 1-4).

* The MS raw datasets of Supplementary Figure 5c are now available for the editor and reviewers using Access key (key#: 8737) . URL: <https://repository.jpostdb.org/preview/10559708835e46371d04355>

Reviewer #1 (Comments for the Author):

This is a well-designed and important study that provides one of the first recent advances in some time on the unique angiogenic pathogenesis of Bartonella. The authors started from scratch by generating a transposon library in *B. henselae* and screening for loss of the proliferative activity for primary endothelial cells (HUVECs). They identified a novel trimeric autotransporter adhesin (BafA) as a factor that could induce HUVECs proliferation that is dependent upon VEGFR2 signaling. The manuscript is well-written and the experiments and resulting data are clearly presented.

Thank you for reviewing our manuscript and giving useful advises. We are encouraged by your positive comments.

The authors should include evidence that the overall growth rate and fitness of the transposons mutants was not altered compared to the wild type as this would also impact cell proliferation.

Thank you for the suggestion. As the reviewer pointed out, various clones with different growth rates were obtained in the transposon library. We have checked the growth rate of clone 623-125 and 804-29 and compared them with the WT strain. Therefore, these results were added to Supplementary figure 4 in the revised version, and the information about them were also added to the appropriate positions in “Identification of BafA” section of Results (lines 108-115).

Additionally, since another trimeric autotransporter adhesin of *B. henselae* (BadA) has been reported to possess pro-angiogenic activity, this should be included in the discussion and compared to BafA.

This comment may contain a misunderstanding by the reviewer. The reviewer seems to be under the impression that BafA is a “trimeric autotransporter adhesin (TAA)”, but it remains unclear whether BafA will belong to the class of TAA. Actually, the passenger domains of TAA like BadA or YadA (*Yersinia* adhesin A) have a coiled-coil long repetitive neck-stalk module, which is necessary for trimeric formation (Szczesny et al. DOI: 10.1371/journal.ppat.1000119, Hoiczky et al. DOI: 10.1093/emboj/19.22.5989). On the other hand, the passenger domain of BafA is predicted to only consist of β -sheet structure (we added the predicted secondary structure to Supplementary figure 8 in the revised version) and is not expected to have the neck-stalk module. From these obvious differences, we speculate that BafA does not form a trimeric state like BadA. Of course, further structural analyses are needed to clarify whether BafA could multimerize, and we consider that it is an important future work. However, we agree that pro-angiogenic activity of BadA needs to be described and compared to BafA, since BafA is indeed an autotransporter protein and has domains homologous to other Gram-negative bacterial adhesins. Thus, their description was added in the appropriate section in Discussion (lines 271-281).

- Line 64: “inject” should read “injects”

The word was replaced accordingly (line 64).

- Lines 69-70: The authors should cite a previous paper showing that VEGF production relies on the presence of BadA. BadA forms a pili-like appendage on the surface of the cell but is not technically a pilus.

The previous papers were cited, and the words were changed accordingly (line 70-71).

- Lines 75-76: Is the mitogenic activity specific for endothelial cells?

We have tried several cell lines to evaluate the mitogenic activity of BafA-PD, among which the activity was confirmed to be specific to endothelial cells. The data on HeLa 229, CHO, and MRC-5 (human fibroblast cell line) are shown in Appendix 1. These data support the specificity of BafA to endothelial cells, but is not considered essential for this paper. Therefore, we do not add it to the paper.

Appendix 1

The cells were treated with PBS (vehicle), bFGF (20 ng/mL), VEGF (20 ng/mL), or BafA-PD (100 ng/mL) and incubated for 3 days. After treatment, cell numbers were counted.

Lines 138-161 Could native (not-tagged) BafA-PD be detected in culture supernatants?

The amount of BafA-PD released from WT strain into the culture medium was predicted to be under the limit of detection, and in addition, the abundant serum proteins might interfere with the detection of BafA. Therefore, we tried to detect native BafA-PD in the fraction eluted from the cation-exchange chromatography to which the culture supernatants were applied. As shown in Appendix 2, we have successfully detected the native BafA-PD from the culture supernatants. However, the fact that BafA is secreted extracellularly has been proved by multiple pieces of evidence in our manuscript, therefore these data were not added to the paper.

Appendix 2

HUVECs were cultured with or without *B. henselae* ATCC 49882 in M199/10% FCS for 24 h. The culture supernatants were collected, dialyzed against 25 mM Tris-HCl, pH 7.5, 50 mM NaCl, and then applied onto HiTrap SP column. The eluate fractions were concentrated with membrane ultrafiltration, and then subjected to immunoblotting using anti-BafA polyclonal antibody.

Lines 243-4: The authors may wish to also mention that *B. bacilliformis* was not observed to have the VirB type IV secretion system genes encoded in the genome. However, *B. bacilliformis* induces clinical angiogenesis - verruga peruana. Thus clinically severe angiogenesis resulting from Bartonella infection does not appear to require the type IV secretion system or its effectors.

Thank you for the helpful advice. As the reviewer suggested, the information about the absence of VirB T4SS in *B. bacilliformis* was added to lines 267-271 in the revised version.

Lines 245-9: The authors should compare BafA and BadA, a protein that has been previously reported to have pro-angiogenic properties. Since both proteins are trimeric autotransporter adhesions, is there any amino acid sequence similarity between the two proteins? Are there any common domains, Is there any structural similarity? Similarly, the BadA homologs in *B. quintana* (Vomps) should also be included in this discussion.

As mentioned above, it is still unclear whether BafA is a trimeric autotransporter adhesin like BadA. To avoid such misunderstanding by readers, we clearly described the difference between BafA and BadA (or Vomps) in the appropriate section in Discussion (lines 308-320).

Line 341: Was the rabbit a specific strain (New Zealand) of white rabbit?

The rabbit strain was JW (Japan White) (Kitayama Labes Co., Ltd.). The strain was specified in lines 420-421 in the revised version.

Line 346: Have the authors tested their *B. henselae* Houston-1 strain to determine if it expresses a functional *badA* gene? Some isolates have been reported to have a truncated version that might have been missed in the screen for cell proliferation. Additionally, repeated subculturing is thought to result in loss of *badA* expression.

Thank you for the useful information. Unfortunately, we could not perform the generally used Western blotting to check the *badA* expression, because we do not have any anti-BadA antibodies, and what is more, it is now also difficult to ask other researchers to provide the antibody due to the COVID-19 situation. As an alternative, we reanalyzed the previously obtained nano-LC-MS/MS data on the total proteins of whole cell lysate extracted from *B. henselae* Houston-1 strain, and as a result, a sufficient number of BadA-derived specific peptides were detected (Appendix 3). This result strongly suggests that *B. henselae* strain we used expresses *badA*. The word “BadA-positive” was added accordingly (line 426). However, we think that these data are not considered essential for this paper. Therefore, we did not add it to the paper.

Figure 1f and Figure 2a are similar and redundant. Deletion of Figure 2a is recommended.

The Figure 2a was deleted accordingly.

Appendix 3

List of BadA- or T4SS-related peptides detected in whole cell lysate of *B. henselae* ATCC 49882 by nano-LC-MS/MS

Accession	Gene	Description	Replicate 1			Replicate 2			Replicate 3		
			# Peptides	# PSMs ^a	Coverage (%)	# Peptides	# PSMs	Coverage (%)	# Peptides	# PSMs	Coverage (%)
A0A0H3LVF3	BH01490 (badA1)	Surface protein/Bartonella adhesin	31	32	28.7	31	37	28.1	32	33	28.1
A0A0H3LX70	BH01510 (badA1)	Surface protein/Bartonella adhesin	10	11	8.3	17	20	16.3	7	7	5.1
Q9R2W4	BH13280 (virB4)	Type IV secretion system protein virB4	0	0	0	1	1	2.3	0	0	0
A0A0H3LXV5	BH13380 (virD4)	Type IV secretion system-coupling protein VirD4	0	0	0	0	0	0	1	1	1.5

^aPSM, peptide spectral matches, filtered at a 1% false discovery rate.

The bacterial cells were harvested from a Columbia agar plate. MS samples were prepared with Pierce Mass Spec Sample Prep Kit for Cultured Cells (Thermo Scientific) according to the manufacturer's instructions. The nano-LC-MS/MS analyses were performed as described in Supplementary Methods. The MS raw datasets have been deposited in the ProteomeXchange Consortium via the jPOST3 partner repository under data-set identifiers PXD018354 (<https://repository.jpostdb.org/preview/1142232375e8806acd3660>, Access key: 9317).

Reviewer #2 (Comments for the Author):

Tsukamoto et al., identify the Bartonella autotransporter BafA as a causative agent of cat scratch disease and angiomas. Using a high content screen for endothelial proliferation after infection with transposon-based random mutants of *B. henselae*, an autotransporter, denoted BafA (Bartonella angiogenesis factor A) was identified as the causative agent. The authors go on to examine BafA gene regulation which clusters in GO terms such as “Angiogenesis”. BafA induces endothelial cell responses in vitro, and endothelial cell outgrowth in aortic ring explants. Matrigel plugs with BafA display increased levels of hemoglobin. Signal transduction analyses show activation of VEGFR2 and Erk1/2 signaling by BafA with close to the same efficiency as for VEGFA. Finally, the authors demonstrate that *B. Quintana* harbors a BafA homologue that stimulates endothelial cell proliferation. This is a careful and ambitious paper with convincing conclusions, which is likely to settle the question on the long-sought identity of causative agent for *B. henselae*-dependent angiomas.

Thank you for reviewing our manuscript and giving helpful advice. We also appreciate your generally positive comments on our work.

General points

1. Please show effects of BafA on endothelial cells in the presence of anti-VEGFA neutralizing antibodies.

Thank you for the suggestion. We agree we should show effects of BafA on endothelial cells in the presence of anti-VEGFA neutralizing antibody (and also anti-VEGFR2 antibody). We have added Supplementary Figure 7 in the revised version, which includes data from an additional experiment done at the reviewer's suggestion. As expected, the presence of anti-VEGF antibody had no effect on BafA-induced cell proliferation, while anti-VEGFR2 antibody remarkably inhibited mitogenic activity of BafA. These results strongly indicate that BafA represents a novel VEGFR2 ligand of which antigenic property is quite different from VEGFA. These findings were added to the “BafA induces angiogenesis through VEGFR2-ERK signaling pathways” section of Results (lines 228-232).

2. What is the affinity of BafA for binding to VEGFR2?

3. Does BafA compete with VEGFA for binding to VEGFR2?

Thank you for the questions. As the reviewer pointed out, we recognize the importance of investigating BafA's affinity and/or competition with VEGFA for binding to VEGFR2 in order to determine the biochemical properties of BafA. However, in this paper, we are focusing on the essential role of BafA

on the formation of vasoproliferative lesion in *Bartonella* infection, and consequently, we demonstrated that BafA accelerates the cell proliferation via VEGFR2 signaling pathway and can be a major factor in angiogenesis. In this context, we consider that these issues indicated by the reviewer are beyond the scope of the present study. We are now working on identifying the receptor binding site in BafA and examining whether the VEGF-binding to VEGFR2 can be prevented by BafA-derivatives. Furthermore, our newly added data (Supplementary Figure 7 in the revised version) strongly suggest that BafA also recognizes the VEGF-binding site in the extracellular domain of VEGFR2 based on the observation that the anti-VEGFR2 antibody inhibited the action of BafA. We would like to ask for your understanding.

4. Does a *B. henselae* BafA mutant fail to induce angiomas?

Unfortunately, no suitable animal models to mimic *Bartonella* infection have been established so far. Therefore, it is difficult at this time to examine if *B. henselae* BafA mutant fails to induce angiomas. Instead of the animal experiments, we have performed the additional experiments of *in vitro* infection according to the reviewer #3's comment and added the data to Supplementary Figure 6 in the revised manuscript. Furthermore, this information was added to the "BafA induces angiogenesis through VEGFR2-ERK signaling pathways" section of Results (lines 205-210). In this cell-based infection model, BafA mutant failed to lead to VEGFR2 phosphorylation induced by the infection of WT or BafA-complement strain. We recognize that there is a gap between cell-based infections and angiomas caused by bacterial infection, but we believe our new data could fill part of the gap.

Minor points

5. Please cite the paper by Riess et al.

Bartonella adhesin a mediates a proangiogenic host cell response.

Riess T, Andersson SG, Lupas A, Schaller M, Schäfer A, Kyme P, Martin J, Wälzlein JH, Eehalt U, Lindroos H, Schirle M, Nordheim A, Autenrieth IB, Kempf VA.

J Exp Med. 2004 Nov 15;200(10):1267-78

The indicated reference was cited as suggested (lines 71, 275, and 310).

6. Does BafA-induced gene regulation support induction of hypoxia-dependent expression pattern in *B. henselae* infected cells as described by Riess et al? Does BafA significantly induce expression of VEGFA? This should be clearly stated.

This comment may be attributed to some misunderstanding by the reviewer. Previously, Riess et al. described that BadA (Bartonella adhesin A)-dependent HIF-1 activation promotes the secretion of pro-angiogenic cytokines from "HeLa cells". However, as far as we know, none of the data using HUVECs has indicated that *B. henselae* enhance the production of VEGF. In fact, Kempf et al. also described *B. henselae* triggers VEGF production of "cell line" EA.hy 926 but not of HUVECs (DOI: 10.1046/j.1462-5822.2001.00144.x). To confirm these previous observations, we also actually tried to detect VEGFA from the culture supernatant of *B. henselae*-infected or BafA-treated HUVECs. As shown in the original manuscript, we were not able to detect VEGF secretion from either infected or BafA-stimulated HUVECs (Appendix 4). These data also support that BafA directly acts as a VEGFR2 agonist, but they are not considered essential for this paper. Also, considering BafA functions via binding to VEGFR2 that is specifically and abundantly expressed on endothelial cell surface, whether HeLa cells respond to BafA would be beyond the scope of this paper. We would like to ask for your understanding.

Appendix 4

Detection of VEGFA from the culture supernatant of HUVEC

VEGFA (pg/mL)					
Treatment			B. henselae-infection		
PBS	DFO	BafA-PD	WT	623-125	623-125/pBAF
N.D. ^a	14.7 ± 0.57	N.D.	N.D.	N.D.	N.D.

^aN.D., not detectable

HUVECs were treated with PBS, Deferoxamine (DFO, positive control), or BafA-PD (100 ng/mL), or infected with Bh (WT, 623-125, or 623-125/pBAF) at MOI of 500 in M199/10% FCS for 48 h. VEGF concentration in culture medium was measured using Quantikine ELISA Human VEGF Immunoassay (R&D Systems) (n=3, biological replicates).

7. In the abstract, the authors state that BafA upregulates VEGFR2 which makes the impression that VEGFR2 expression levels are upregulated. However, the authors intention is likely to point out that BafA induces VEGFR2 signal transduction? Please rephrase.

Thank you for the suggestion. The indicated point was rephrased properly (lines 39-41).

Reviewer #3 (Comments for the Author):

The study by Tsukamoto et al., presents the discovery and characterization of a novel pro-angiogenic factor secreted by the zoonotic pathogen *Bartonella*, which acts as an agonist of the VEGF-receptor 2 (VEGFR2). With an exhaustive set of experiments, the authors unambiguously demonstrate that the passenger domain of a bacterial autotransporter (named here BafA) is sufficient to trigger cell proliferation, eventually resulting in angiogenesis. Their data indicate that the passenger domain of BafA (BafA-PD), which is cleaved after translocation through the outer-membrane, constitute a novel secreted effector that is likely to play an important role in host colonization and/or adaptation of this stealthy pathogen. At the molecular level, the authors show that BafA-PD interacts with VEGFR2 and triggers downstream signaling, thus acting as a VEGF-like molecule. Combining simple and complex in-vitro models, infections and use of recombinant proteins, the authors provide a very solid body of evidence, further confirmed using an elegant in-vivo assay. This study is definitely an important contribution for the understanding of the pathogenesis of the pathogen *Bartonella*. Indeed, although the mitogenic properties of *B. henselae* have long been recognized, the underlying factor(s) remained elusive. Moreover, the description of a bacterial VEGF-like secreted effector as a novel strategy for a pathogen to establish its infection niche will very likely catch the interest of a broad scientific audience.

Thank you for reviewing our manuscript and giving valuable information. Your positive comments really encourage us.

Minor points

-1. (General) One general issue I have on the overall report is a possible gap between the data on cell proliferation and those using angiogenesis as a read-out. On one hand, the authors present very convincing angiogenesis assays with purified BafA-PD. On the other hand, infection and cell signaling data only used cell proliferation as a read-out. Although it is clear that cell proliferation can be used to some extent as a proxy for angiogenesis, the two processes are not fully congruent. To overcome that possible gap, I would suggest that the authors provide at least one infection experiment (with wild-type, BafA mutant and complemented strain) while assessing phosphorylation of the VEGF receptor-2 and the phosphorylation of ERK1/2. This would fully fill the gap. Alternatively, the conclusion could be somehow tuned to account for this gap. Further, it would be interesting to test whether the isolated BafA mutants are still able to induce VEGF secretion by HUVECs upon infection.

Thank you for the suggestion. As we have already described in the response to reviewer #2, we performed an additional cell-based infection experiment, and the data were added to Supplementary Figure 6 in the revised version. In addition, the explanation about it was also added to the “BafA induces

angiogenesis through VEGFR2-ERK signaling pathways” section of Results (lines 205-210). As expected, the BafA mutant failed to lead to VEGFR2 phosphorylation otherwise induced by infection by WT or BafA-complement strain. We believe that our additional data would fill the gap indicated by the reviewer. Regarding VEGF secretion by HUVECs, as described above, HUVECs do not appear to enhance VEGF production in response to *B. henselae* infection or BafA stimulation (Appendix 4 and Kempf et al. DOI: 10.1046/j.1462-5822.2001.00144.x). We consider this observation further supports that BafA itself possesses VEGF-like activity.

2. (General) The authors have used the *B. henselae* strain ATCC 49882 for their experiments. Importantly, it has been reported that at least two variants of the strain exist, likely as the result of in-vitro passaging (DOI: 10.1111/cmi.12070). One variant (var-2) expresses the trimeric autotransporter BadA but fail to express the VirB/D4 T4SS and to translocate T4SS effector in infected cells. This strain however strongly induce VEGF expression when infecting HUVECs. The other variant (var-1) displays symmetrically opposite properties (BadA negative, T4SS and effector translocation positive, no induction of VEGF). This has caused discrepancies in results obtained between different laboratories, and could be the underlying reason for the points raised in comments # 6 and #8. It would therefore be important to clarify the properties of the strain used for this study (BadA expression, T4SS expression, VEGF induction upon infection). Note that this point does not questions the pertinence of the BafA-related phenotype, but may be highly relevant for the interpretations of the global infection strategy of the studied pathogen.

Thank you for giving us the detail information about the *B. henselae* variants and helpful advices. Since reviewer #1 also pointed out the same issue, we attached the nano-LC-MS/MS data which we used to examine the expression of BadA and VirB/D4 T4SS (Appendix 2). In this analysis, a large number of BadA-derived peptides were detected, while few T4SS-derived peptides including its effectors were detected. Although the MS analysis is not considered to be quantitative, these results strongly suggest that the bacterial strain we used may be close to variant 2 (BadA-positive variant) and has not lost BadA expression during the subculturing process. As for VEGF expression induced by *B. henselae* infection, the authors of the paper indicated by the reviewer (Lu et al. DOI: 10.1111/cmi.12070), like other reports, also used HeLa cells, but not HUVECs, to evaluate VEGF expression. The reviewer may have missed this specific point. As the reviewer suggested, we agree that it is necessary to determine VEGF production from other cells surrounding vascular endothelial cells (paracrine loop) when we discuss the overall infection strategy of the bacteria. However, evaluating the VEGF secretion from HeLa cells is not directly related to the activity of BafA, and we do not believe that description of the issue is essential for this paper. We would like to ask for your understanding.

3. (Line 69-70): Could the authors clarify the term “ pili” used in the sentence “ ” The VEGF production

from the cells relies on the presence of *B. henselae* pili” ? Do they refer to the trimeric autotransporter BadA?

Thank you for pointing out misuse of the term. This point was also indicated by reviewer #1, and the words were already replaced properly (lines 70-71).

4. (Lines 119-124). Could the authors indicate in the main text that data were generated by INDIRECT co-culture and that the statistically significant expression changes observed result from the comparison to uninfected cells? It is clearly stated in the figure legend, but adding it to the main text would increase clarity.

Thank you for the suggestion. Clarifications were made in the text accordingly (lines 128-132).

5. Regarding the RNA-seq analysis: could the authors provide a (supplementary) table with the genes displaying differential expression (presented in fig. 2) – including the normalized expression values for the three tested conditions?

Thank you for the suggestion. As suggested by the reviewer, we provided the table with differentially expressed genes including the normalized expression values (RPKM: reads per kilobase of transcript per million mapped reads) of three replicates. These data are now included in Fig. 2e in the Source Data file.

The Venn diagram presented in Fig. 2b actually only show a rather modest overlap between the wild-type and the complemented strain. Could this be due to the stringent statistical cut-off applied?

We do not know exactly why the Venn diagram shows only a modest overlap between the wild-type and the complemented strains. Despite the use of common cut-off values ($FDR < 0.05$, $|Fold\ change| > 1.5$) in our analysis, differentially expressed genes resulting from co-culturing with *B. henselae* were not detected so often. One possibility is that in the INDIRECT co-culture condition, a small number of factors that affect HUVEC (including BafA) could be secreted by *B. henselae*.

A few word on the genes up-regulated in both wild-type and mutant strain could also be of general interest.

As expected by the reviewers, in fact, the GO analysis of genes regulated in both WT and mutant strains

enriched the term “Angiogenesis”. However, we think it is better not to add the data to Figure 2, because it would become redundant. We would like to ask for your understanding.

Further, as the authors discovered that BafA-PD acts as a VEGF-like molecule, a comparison to the transcriptional response of HUVECs to VEGF treatment (from literature) would be appealing for the interpretation of the observed changes.

As the reviewer pointed out, it is of interest to compare the transcriptional response of HUVECs to BafA treatment with that to VEGF treatment. When we actually compared our data with the data reported in the literature (Liu et al. DOI: 10.1007/s12031-015-0653-z), only a few DEGs (CCL2, CXCLR4, and MMP1) were common but most of them were not. Moreover, since not all of 154 DEGs were specified in this literature, it was not possible to accurately compare both DEGs. In order to discuss this issue accurately, we need to re-analyze this using HUVECs treated with BafA-PD, which we think would be beyond the scope in this paper. We would like to ask for your understanding.

6. The authors convincingly demonstrate that BafA-PD interacts with the VEGF receptor-2, acting as an agonist. They clearly show that BafA-PD stimulates cell proliferation together with the activation of phosphorylation of VEGFR2 and ERK1/2. Interestingly, a previous report (DOI: 10.1111/j.1462-5822.2010.01545.x) assessed the effect of *B. henselae* (var-1) on VEGF signaling using infection of HUVECs and concluded that the pathogen actually inhibits VEGFR2 signaling (in VEGF-stimulated cells) in a T4SS-dependent manner. The authors showed that the same phosphorylation site activated by BafA-PD (Tyr 1175) is actually inhibited by their wild-type strain. This study should be cited and the discrepancies discussed (likely linked to point #2)

Thank you for providing this valuable information. As suggested, the indicated reference was cited (lines 327), and the discrepancies are discussed in Discussion (lines 325-340).

7. The authors propose BafA as a factor contributing to host adaptation (line 45). In that context, the authors could consider the following point for discussion. As stated in the introduction, vasoproliferative lesions seem to be restricted to the infection of human as an incidental host, and this mainly in case of immunosuppression (*B. henselae* and *B. elizabethae*). The most striking case of vasoproliferation described to date is the case of human infection by *B. bacilliformis* (verruca peruana). It would make sense to include this species in the tree of BafA-homologues presented in fig. 5g.

Thank you for the suggestion. As the reviewer pointed out, *B. bacilliformis* were added to the tree of BafA-homologues in Figure 5g, and the suggested information was incorporated in Discussion (lines

359-364) and Materials and Methods (lines 666-667).

The lack of report for vasoproliferative lesion in natural host (with the exception of *B. bacilliformis*) does not exclude angiogenesis is a key requirement for the establishment of *Bartonella* infection in its natural host. Notably, the endothelium has already long been proposed as the putative replicative niche from which the bacteria invade the erythrocytes (blood seeding niche). That point could be addressed in the discussion.

Thank you for the suggestion. The suggested points were added to Discussion (lines 364-371)

8. Although the data presented unambiguously support the role of BafA in angiogenesis, this role was not tested in any infection model (only cell proliferation was). Intriguingly, a previous publication by Scheidegger et al., (DOI: 10.1111/j.1462-5822.2009.01313.x), which used infection of HUVECs-derived spheroids and sprouting as readout, concluded that angiogenesis was strongly promoted the VirB/D4 type IV secretion (T4SS) and the secreted effector BepA. However, they only observed “basal angiogenesis” in the absence of the T4SS. This is in clear contrast to experiments addressing cell proliferation, a process that was shown to be T4SS-independent. The study of Scheidegger et al., should be cited and the discrepancies briefly discussed (likely linked to the point raised in comment #2). It sure would be interesting to see the effect of a BafA mutant in an infection experiment with angiogenesis as read-out, but this may go beyond the scope of the present report.

Thank you for pointing us to this intriguing previous study. As mentioned above, we added data from cell-based infection experiment. Moreover, the indicated reference was cited (line 265), and necessary information was added (lines 261-267).

In the indicated reference, the authors also used *B. henselae* var-1 as the wild-type strain, and therefore, it would be unpredictable whether similar results could be obtained even if we perform the same experiment for the purpose of readout of BafA-triggered angiogenesis. However, there is no doubt that experiment using HUVEC spheroids would be a valuable infection model for evaluating *Bartonella*-induced angiogenesis. Moving forward, we will try to examine the effects of BafA in infection experiments including the spheroids model or that beyond cell-base.

REVIEWERS' COMMENTS:

Reviewer #1 (Remarks to the Author):

This revised manuscript is very well written and addresses all of my concerns. The authors are to be congratulated for an excellent and thorough study of this newly characterized BafA protein. There are a few minor grammatical errors, mostly in the revised text, that should be addressed before publication.

Burt Anderson

Reviewer #2 (Remarks to the Author):

The authors have responded in a satisfactory manner. It is a pity that in vivo models for Bartonella infection are lacking; such in vivo data would have increased the general interest and impact of the study. However, the added data for example on the loss of VEGFR2 phosphorylation by BafA in the presence of anti-VEGFR2 antibodies directed towards the VEGFA-binding site on the receptor, provides important information.

Reviewer #3 (Remarks to the Author):

[No further comments for authors]

Point-by-Point Responses to Reviewers' comments

We are most grateful that all reviewers regard our revised manuscript as suitable, in principle, for publication in Nature Communications. Detailed point-by-point responses to the reviewers' comments are provided below.

*The comments provided by the reviewers are in blue italics. Our responses are shown in black roman type.

REVIEWERS' COMMENTS:

Reviewer #1 (Remarks to the Author):

This revised manuscript is very well written and addresses all of my concerns. The authors are to be congratulated for an excellent and thorough study of this newly characterized BafA protein. There are a few minor grammatical errors, mostly in the revised text, that should be addressed before publication.

We appreciate you reviewing our manuscript, and we are very delighted that our revised manuscript has addressed all of your concerns. We have carefully checked the manuscript and revised a few outstanding minor grammatical errors.

Reviewer #2 (Remarks to the Author):

The authors have responded in a satisfactory manner. It is a pity that in vivo models for Bartonella infection are lacking; such in vivo data would have increased the general interest and impact of the study. However, the added data for example on the loss of VEGFR2 phosphorylation by BafA in the presence of anti-VEGFR2 antibodies directed towards the VEGFA-binding site on the receptor, provides important information.

Thank you for reviewing our manuscript. It is a little unfortunate for us that in vivo models for *Bartonella* infection are lacking, but we would like to continue working toward solving these issues in our future work. As for the experiments with anti-VEGFR2, we are thankful to your comments that allowed us to add more compelling data.

Reviewer #3 (Remarks to the Author):

[No further comments for authors]